# How Closely Does Induced Agarwood’s Biological Activity Resemble That of Wild Agarwood?

**DOI:** 10.3390/molecules28072922

**Published:** 2023-03-24

**Authors:** Sheng Ma, Manqin Huang, Yunlin Fu, Mengji Qiao, Yingjian Li

**Affiliations:** Key Laboratory of National Forestry and Grassland Administration on Cultivation of Fast-Growing Timber in Central South China, College of Forestry, Guangxi University, Nanning 540004, China

**Keywords:** induced agarwood, wild agarwood, components, antioxidant activity, anti-acetylcholinesterase activity, anti-α-glucosidase activity

## Abstract

Continuous innovation in artificially-induced agarwood technology is increasing the amount of agarwood and substantially alleviating shortages. Agarwood is widely utilized in perfumes and fragrances; however, it is unclear whether the overall pharmacological activity of induced agarwood can replace wild agarwood for medicinal use. In this study, the volatile components, total chromone content, and the differences in the overall activities of wild agarwood and induced agarwood, including the antioxidant, anti-acetylcholinesterase, and anti-glucosidase activity were all determined. The results indicated that both induced and wild agarwood’s chemical makeup contains sesquiterpenes and 2-(2-phenylethyl)chromones. The total chromone content in generated agarwood can reach 82.96% of that in wild agarwood. Induced agarwood scavenged 1,1-diphenyl-2-picrylhydrazyl (DPPH) radicals and 2,2′-azino-bis (3-ethylbenzothiazoline-6-sulfonic acid) (ABTS^+^) radicals and inhibited acetylcholinesterase activity and α-glucosidase activity with IC_50_ values of 0.1873 mg/mL, 0.0602 mg/mL, 0.0493 mg/mL, and 0.2119 mg/mL, respectively, reaching 80.89%, 93.52%, 93.52%, and 69.47% of that of wild agarwood, respectively. Accordingly, the results distinguished that induced agarwood has the potential to replace wild agarwood in future for use in medicine because it has a similar chemical makeup to wild agarwood and has comparable antioxidant, anti-acetylcholinesterase, and anti-glucosidase capabilities.

## 1. Introduction

*Aquilaria*, belonging to the family Thymelaeaceae, is renowned for producing agarwood. According to documents, the history of the human use of agarwood can be traced back more than 2000 years. The genus *Aquilaria* comprises 21 species that are mainly distributed through India, Myanmar, Laos, Vietnam, Cambodia, Malaysia, Sumatra, Borneo, Philippines, Papua New Guinea, and other countries [1]. The formation of agarwood occurs slowly and infrequently in nature, only around 7 to 10%; hence, premium agarwood can cost up to $10,000 (USD) per kg [2,3]. Due to its extensive use in fragrance, incense, cosmetics, religion, and medicine, agarwood has grown in popularity and constitutes a multibillion-dollar market [4]. However, the existing availability of wild agarwood makes it difficult to meet the market’s rising demand. Furthermore, overharvesting and a lack of environmental protection are causing a slow decline in the availability of wild *Aquilaria* resources. In 2005, all extant species of *Aquilaria* were listed in Appendix II of CITES (Convention on International Trade in Endangered Species of Wild Fauna and Flora), resulting in a greater scarcity of natural agarwood, which also led to a continued increase in the price of wild agarwood and it becoming a collector’s item. Moreover, in recent years, Chinese patent medicines based on traditional Chinese medicine have received considerable attention, and in accordance with data from the China Food and Drug Administration, there are currently over 100 different varieties of Chinese patent medicines that use wild agarwood as a raw material, such as ChenXiangHuaQiWan. Likewise, the growth of agarwood’s medical uses is being hampered by the scarcity of wild agarwood. China has recently seen a remarkable increase in the areas planted with *Aquilaria* trees. According to published literature, 5285 hectares of *Aquilaria* trees, including 5245 hectares of *Aquilaria sinensis*, were newly planted in China between 2006 and 2010 [5]. Currently, following incomplete statistics, China has more than 30,000 hectares of *Aquilaria* trees, and physical, chemical, and biological techniques have been devised to speed up the creation of agarwood in *Aquilaria* plantations [6], effectively increasing the number of agarwood trees and easing the demand for agarwood in the fragrance and cosmetics industry. Recent studies have identified a number of pharmacological actions of induced agarwood, including antioxidant, anti-acetylcholinesterase, anti-glucosidase, and anti-inflammatory properties [7,8,9,10]. Unfortunately, it is currently unknown if induced agarwood may be used in place of wild agarwood for the manufacturing of certain Chinese patent medicines due to the paucity of pertinent studies on the distinction between the general activity of the two types of agarwood.

Previously, our group successfully induced the formation of agarwood by inoculating *A. sinensis* with fungal inducers [11]. In this study, the chemical composition of induced agarwood and wild agarwood was analyzed using gas chromatography-mass spectrometry (GC-MS) and ultraviolet and visible spectrophotometry (UV-Vis), and the difference in overall activity between the two was also determined, including antioxidant activity (1,1-diphenyl-2-picrylhydrazyl (DPPH) and 2,2′-azino-bis (3-ethylbenzothiazoline-6-sulfonic acid) (ABTS^+^) radical clearance ability), anti-acetylcholinesterase activity, and anti-alpha-glucosidase activity, through which the pharmacological effects of induced agarwood can be evaluated and provide a basis for its utilization in Chinese patent medicines. 

## 2. Results

### 2.1. Chemical Composition Analysis

#### 2.1.1. Volatile Chemical Composition Analysis

The total ion flow chromatogram for IA and WA are displayed in Figure 1. In total, 36 compounds were detected and tentatively identified from IA, with a total relative content of 96.01% (Appendix A), of which 16 sesquiterpenes and six 2-(2-phenylethyl)chromones were identified. IA had a 30.12% overall relative sesquiterpenoid content, with the majority of these sesquiterpenoids being 5,8-Dihydroxy-4a-methyl-4,4a,4b,5,6,7,8,8a,9,10-decahydro-2(3H)-phenanthrenone (18.56%), humulene epoxide I (0.55%), valerenic acid (1.64%), calarene epoxide (0.38%), arctiol (0.69%), and other sesquiterpenoids. IA contained six 2-(2-phenylethyl)chromones with a combined relative concentration of 48.76%. The following were among them: 2-(2-phenylethyl)chromone (9.46%), 7-methoxy-3-(p-methoxyphenyl)chromone (3.06%), 8-methoxy-2-(2-phenylethyl)chromone (8.39%), 6-methoxy-2-(4-methoxyphenethyl)chromone (2.15%), 6,7-dimethoxy-2-(2-phenylethyl)chromone (23.17%), and 6,7-dimethoxy-2-(4-methoxyphenethyl)chromone (2.53%). In combination with this, IA has also been shown to contain trace levels of the triterpenoid squalene and the aromatic compound benzylacetone (Table 1).

With a combined relative level of 92.34%, fifty chemical components, primarily sesquiterpenes and aromatic components were found in the WA extracts. There were 21 sesquiterpenes in WA, with a calculated relative content of 37.60%, mainly including 2,4,6,7,8,8a-hexahydro-3,8-dimethyl-4-(1-methylethylidene)-(8S-cis)-5(1H)-azulenone (9.90%), Acorenone B (2.10%), and valerenic acid (1.64%). Aromatic substances and their derivatives were also the main volatile components of WA, such as 2,6-bis(1,1-dimethylethyl)-4-methyl-,methylcarbamate-phenol (8.11%), 2,2’-methylenebis [6-(1,1-dimethylethyl)-4-methyl-phenol (4.54%) (Table 1), and various aromatic substances in the Appendix A. Apart from these, 2-(2-phenethyl)chromones were found only as 6,7-Dimethoxy-2-(2-phenethyl)chromone and a triterpenoid supraene in WA.

According to Table 1, IA and WA had five common components, including three sesquiterpenes (valerenic acid, 5,8-Dihydroxy-4a-methyl-4,4a,4b,5,6,7,8),8a,9,10-decahydro-2(3H)-phenanthrenone, and 2aS, 3aR,5aS,9bR)-2a,5a,9-Trimethyl-2a,4,5,5a,6,7,8,9b-octahydro-2H-naphtho[1,2-b]oxireno[2,3-c]furan), and a chromone (6,7-Dimethoxy-2-(2- phenethyl)chromone).

#### 2.1.2. Total Chromone Content

The standard curve’s equation, Y = 0.1071X − 0.0332, R^2^ = 999, showed acceptable accuracy at concentrations between 1 and 6 mg/mL for determining the total chromone content. According to the measurements, WA and IA have total chromone contents of 7.83 ± 0.54% and 6.50 ± 0.40%, respectively, although there was a significant difference between the total chromone content of IA and WA (*p* < 0.05). However, the total chromone content in IA can reach 82.96% of that in WA (Figure 2). Obviously, it was revealed that induced and wild agarwood had a similar chemical composition.

### 2.2. Bioactivity Analysis

#### 2.2.1. Antioxidant Activity Analysis

DPPH free radical scavenging capacity

As in Figure 2, at a mass concentration of 0.1 mg/mL, the IA and WA extract had some scavenging capacity for DPPH radicals but the scavenging rates were weaker than that of VC. Afterwards, the clearance rate of the DPPH free radicals tended to rise with rising extract concentrations, displaying some dose dependence (Figure 3a).

The IC_50_ values for DPPH free radical scavenging by IA and WA were 0.1873 mg/mL, and 0.1515 mg/mL, respectively (Figure 3b). In this situation, the ability of IA to scavenge DPPH radicals can reach 80.89% of that of WA. Intriguingly, the DPPH radical scavenging rate tended to stabilize at an extract concentration of 0.8 mg/mL, at which point the IA and WA extracts scavenged 91.26 ± 0.60% and 91.59 ± 0.22% of DPPH radicals, respectively, with no significant difference between them (*p* > 0.05). In other words, the findings showed that the agarwood produced by fungal inducers was quite similar to natural agarwood in terms of its capacity to scavenge DPPH free radicals.

ABTS^+^ free radical scavenging capacity

As illustrated in Figure 3, the extracts of IA and WA showed a favorable scavenging effect on ABTS^+^ radicals at a mass concentration of 0.05 mg/mL. In this state, the clearance rates were 39.98 ± 3.16% and 43.47 ± 2.39%, respectively, while the scavenging of DPPH radicals in IA and WA were not as superior as that of VC (Figure 4a). Moreover, the scavenging activity of agarwood also increased with the increase in its mass concentration from 0.1 to 0.4 mg/mL. A considerable concentration–dose dependence was evident in the scavenging of ABTS^+^ as the extract’s mass concentration increased. The IC_50_ values for the ABTS^+^ radicals were 0.0602 mg/mL and 0.0563 mg/mL for the IA and WA extracts, respectively (Figure 4b). In other words, IA was able to scavenge 93.52% of the ABTS^+^ radicals from WA. Furthermore, it was also noteworthy that the scavenging of ABTS radicals by IA and WA was 91.03 ± 1.01% and 94.80 ± 0.85%, respectively, by the mass concentration of extract at 0.2 mg/mL, and that there was no statistically significant difference between their scavenging rates (*p* > 0.05). Therefore, the results revealed that the induced agarwood had an excellent ABTS^+^ free radical cleaning effect and was extremely close to wild agarwood.

#### 2.2.2. Anti-Acetylcholinesterase Activity Capacity

The extracts of WA and IA possessed a limited ability to inhibit acetylcholinesterase activity at an agarwood extract mass concentration of 0.025 mg/mL, at which point IA and WA inhibited acetylcholinesterase by 42.02 ± 1.72% and 47.57 ± 3.80%, respectively, yet not as effectively as tacrine. When the extract concentration was raised from 0.025 mg/mL to 0.4 mg/mL, the inhibition of acetylcholinesterase tended to grow rapidly. In contrast, acetylcholinesterase inhibition increased moderately in IA and WA at extract mass concentrations ranging from 0.4 mg/mL to 0.8 mg/mL (Figure 5a). The IC_50_ values for the inhibition of acetylcholinesterase activity by IA and WA extracts were 0.0493 mg/mL and 0.0350 mg/mL, respectively, and the ability of IA to inhibit acetylcholinesterase activity could approach 70.99% of that of wild agarwood (Figure 5b). However, the inhibition of acetylcholinesterase by IA and WA was, additionally, 80.17 ± 2.72% and 84.06 ± 1.36% at a mass concentration of 0.4 mg/mL, respectively, with no significant difference between these two extracts (*p* > 0.05). The outcomes revealed that the produced agarwood had a beneficial impact on acetylcholinesterase activity inhibition and that the scavenging effect was more comparable to that of wild agarwood. 

#### 2.2.3. Anti-α-Glucosidase Activity Capacity

The inhibitory effect of the IA and WA extracts on α-glucosidase increased with increasing mass concentration, exhibiting a certain dose-dependence, with the inhibitory effect of IA and WA on α-glucosidase activity increasing rapidly at mass concentrations of the extracts from 0.025 to 0.4 mg/mL, with IA inhibition increasing from 11.50 ± 2.69% to 72.56 ± 0.89% and WA inhibition increasing from 18.51 ± 2.90% to 77.25 ± 2.06% (Figure 6a). The capacity of IA to inhibit α-glucosidase activity could reach 69.47% of that of WA, according to the IC_50_ values of 0.2119 mg/mL and 0.1472 mg/mL for IA and WA, respectively (Figure 6b). However, the α-glucosidase activity levelled off when the mass concentration of the agarwood extract was increased to 0.8 mg/mL, and the inhibition rates of α-glucosidase were 85.14 ± 2.59% and 89.19 ± 1.68% for IA and WA, and there was no significant difference in the inhibition rate of α-glucosidase between IA and WA (*p* > 0.05). In summary, the results demonstrated that the induced agarwood had a superb α-glucosidase inhibitory activity, approaching that of wild agarwood.

## 3. Discussion

### 3.1. Chemical Composition of Induced Agarwood and Wild Agarwood

Chen et al. (2011) reported that the chemical composition of *A. sinensis* is dominated by its primary metabolites, such as fatty acids and alkanes [12]. Conversely, in this study, it was evident that alkanes and fatty acids were absent from the agarwood produced by fungal inducers. The reasons for this phenomenon was that with the inoculation of fungal inducers, the fungus released a range of cell wall degrading enzymes. For instance, *Trichoderma*, *Aspergillus*, and *Penicillium* were recognized as models—from bench to industrial scale production—of cellulases that break down plant cell walls to obtain nutrients for their own survival and reproduction [13,14]. Subsequently, the defense mechanisms of the *A. sinensis* tree were engaged in a response to stop further fungal attack, with linked genes controlling the production of a number of endogenous hydrolases by vesicles, resulting in programmed cell death and cell wall lysis [15,16]. At the same time, a series of precursors for terpenoid backbone biosynthesis and terpene synthase were induced to promote terpenoid synthesis, and the phenol propane metabolic pathway was also activated, such as polyketide synthase, to synthesize diarylpentanoid, which is a prerequisite for the synthesis of the chromones that were subsequently modified by a series of enzymes to generate the various types of 2-(2-phenylethyl)chromones [17,18], as sesquiterpenes and chromones are products of plant resistance to adversity and are also a source of pharmacological action. In comparison to the study by Liao et al., on agarwood caused by the physical trauma method, the total relative content of chromones and sesquiterpenes in induced agarwood produced by fungal inducers was superior [19]. The point is, that the sesquiterpene content and quantity of induced agarwood was not only close to that of wild agarwood but the presence of a variety of other shared components, as well as the total chromone content, was also close to that of wild agarwood. Thus, the composition type in this induced agarwood had a greater similarity to that of wild agarwood.

### 3.2. Biological Activity of Induced Agarwood and Wild Agarwood

The body’s overabundance of free radicals can cause aging, cancer, cardiovascular disease, inflammation, and other disorders that are dangerous to human health [20]. Numerous sesquiterpenes and chromones in agarwood have been found to have antioxidant effects, according to research. The 8α-5α.6β.7β-trihydroxy5,6,7,8-tetrahydro-2-[2-(3-hydroxy-4-methoxyphenyl)ethyl]chromone, 6-hydroxy-2-[2-(4-hydroxy-3-methoxyphenyl)vinyl]chromone and 2-oxo-12-hydroxy-hinesol (sesquiterpene) in artificially punched hole agarwood eliminated DPPH radicals quite effectively, demonstrating the antioxidant capabilities of these three substances [21]. β-Caryophyllene is a sesquiterpene with excellent antioxidant potential [22]. In addition, 6-hydroxy-2-[2-(2-hydroxyphenyl)ethyl]chromone, 8-chloro-6-hydroxy-2-[2-(3-methoxy-4-hydroxyphenyl)ethyl]chromone, 6,7-dimethoxy-2-[2-(4-hydroxyphenyl)ethyl]chromone and 6,7-dimethoxy-2-[2-(4-hydroxyphenyl)ethyl]chromone powerfully scavenged ABTS^+^ radicals and also exhibited significant antioxidant properties [7]. The composition of the agarwood produced by this induction, which has rarely been reported in the literature for its antioxidant capacity, combined with the results of the antioxidant test, indicates that this induced agarwood has an excellent antioxidant capacity and the presence of a number of substances that also have antioxidant capacity. Therefore, this agarwood has some relevance for future development and for the application of replacing wild agarwood as an antioxidant.

Alzheimer’s disease (dementia) is a disease that impairs cognitive function and results in brain cell deterioration. One study pointed out that there were about 50 million Alzheimer’s patients worldwide in 2020. More frighteningly, the number is predicted to rise to 152 million by 2050, and the annual cost of treating patients is up to USD 1 trillion, making it a global problem [23]. Tacrine and galantamine, two commonly used clinical treatments for Alzheimer’s disease, have some effect but come with a number of side effects such as drowsiness, exhaustion, gastrointestinal discomfort, elevated transaminase levels, and heart arrhythmias [24,25], which are not conducive to their long-term use by Alzheimer’s patients. It was proved that agarwood contains a number of substances with anti-acetylcholinesterase activity. (-)-7βH-eudesmane-4α,11-diol, (5S,7S,9S,10S)-(+)-9-hydroxy-selina-3,11-dien-12-al and 12,15-dioxo-α-selinene are three sesquiterpenes that have exhibited some anti-acetylcholinesterase activity via inhibition [26]. The inhibition of acetylcholinesterase by 6,8-dihydroxy-2-[2-(4-methoxyphenyl)ethyl]chromone, 6,7-dimethoxy-2-[2-(3-hydroxy-4-methoxyphenyl)ethyl]chromone, 6-hydroxy-7-methoxy-2-[2-(3-hydroxy-4-methoxyphenyl)ethyl]chromone and 6-hydroxy-2-[2-(3-methoxy-4-hydroxyphenyl)ethyl]chromone in punched hole agarwood was more than 12% [27]. An even more exaggerated fact was that (6S,7S,8S)-6,7,8-trihydroxyl-2-(4-hydroxyl-3-methoxylphenylethyl)-5,6,7,8-tetrahydro-4H-chromen-4-one, (6S,7S,8S)-6,7,8-trihydroxyl-2-(3-hydroxyl-4-methoxylphenylethyl)-5,6,7,8-tetrahydro-4H-chromen-4-one and 7-hydroxyl-6-methoxyl-2-[2-(4-hydroxyl-3-methoxyl-phenyl)ethyl]chromone all inhibited acetylcholinesterase by more than 32% [28]. In this research, both the induced agarwood and wild agarwood showed favorable anti-acetylcholinesterase activity. Interestingly, according to the TCMSP database, the acetylcholinesterase-related target, Muscarinic acetylcholine receptor M1, is acted upon by 6,7-dimethoxy-2-(2-phenylethyl)chromone, which was discovered from induced agarwood extracts [29]. The results demonstrated that induced agarwood has the potential for the prevention of, or the development of pharmaceuticals for the treatment of, Alzheimer’s disease, or even that these natural products may avoid the development of side effects during treatment.

According to the statistics of the International Diabetes Federation, in 2021, 536.6 million people were suffering with diabetes worldwide [30]. Unfortunately, nephropathy, macrovascular disease, and microvascular disease are among the many consequences of diabetes that can lead to devastating societal problems [31]. Alpha-glucosidase inhibitors, such as acarbose, can block a variety of α-glucosidase in the human small intestine, delaying the absorption of glucose and reducing post-prandial blood glucose levels in a dose-dependent manner. Regrettably, diabetes is a chronic disease that must be treated with long-term medicine; however, these are prone to problems, such as gastrointestinal, liver and kidney damage [32]. A variety of studies have previously demonstrated the existence of various components from agarwood that can inhibit alpha-glucosidase. 2-[(2β,8α,8aα)-8,8a-dimethyl-1,2,3,4,6,7,8,8a-octahydronaphthalen-2-yl]propane-1,2-diol, eudesma-4-en-8,11-diol and methyl-15-oxo-eudesmane-4,11(13)-dien-12-oate are two sesquiterpenes that have inhibited α-glucosidase efficiently [33]. Twenty-one chromones, including 8-hydroxy-2-[2-(4-methoxyphenyl)ethyl]chromone, 6,7-dihydroxy-2-[2-(4-hydroxy-3-methoxyphenyl)ethyl]chromone, and 6,7-dimethoxy-2-(2-phenylethyl)chromone also revealed a suppressive effect on α-glucosidase activity [9]. Likewise, the associated target (Peroxisome proliferator-activated receptor gamma) of diabetes can be affected by 6,7-dimethoxy-2-(2-phenylethyl)chromone [29]. In accordance with the test results, the induced agarwood and wild agarwood have similar abilities to inhibit α-glucosidase activity, and both have excellent inhibition effects. Thus, the induced agarwood may not only be used to create diabetes medications but also to lessen their negative effects. Even when agarwood is used as an ingredient in traditional Chinese medicine to lower post-prandial blood sugar, the incidence of diabetes is greatly reduced.

## 4. Materials and Methods

### 4.1. Materials

A healthy, 7-year-old, ≥5 cm in diameter *A. sinensis* planted in Pingding Town, Huazhou City, Maoming City, Guangdong Province, China was selected for the inoculation experiment. In the region, the longitude and latitude are 10.422263 and 22.050065, respectively, and the average annual temperature is roughly 23 °C, the average annual precipitation is 1800 mm, and the annual relative humidity is 80%. The inoculation experiments used fungal inducers (containing *Aspergillus penicillioides*, *Gongronella butleri*, *Aspergillus sydowii*, and other fungi) that were obtained by extracting strains of natural agarwood, which were enriched and expanded [11]. Spiral holes were drilled into the *Aquliaria* trees and a high-pressure perfusion device was used to inject the fungal inoculant into the holes at a depth of approximately two-thirds of the tree’s diameter (Figure A1, Appendix C). Three plants were randomly harvested after 11 months of inoculation, dried, and the white wood was removed to obtain a brown-black resin from the induced agarwood (IA) to use as the test samples (Figure 2a and Figure 7). In contrast, an uninoculated *A. sinensis* was harvested without agarwood (Appendix A). Additionally, wild agarwood (WA) with a high oil content produced by *A. sinensis* was purchased as a positive control from the agarwood market in Hainan Province, China (Figure 7).

### 4.2. Methods

#### 4.2.1. Chemical Composition Determination

Volatile chemical composition determination

A total of 25 mL of ether was added separately to 2 g of the samples, followed by sonication in an ice-water bath for 30 min and then filtration. The ether extract was subjected to the aforementioned procedures until it became colorless. Then, the fume hood was evaporated and 3 mL of dichloromethane was taken to dissolve the ether extract, which was filtered through a 0.22 μm organic system filter membrane, and injected into a feed vial for GC-MS (Agilent, 7890B, Santa Clara, CA, USA) analysis. A capillary column made of DB-5MS Ultra Inert and a mass spectrometer 7890B-7000D were installed in the instrument. We used the following temperature program, 90 °C to 160 °C at 55 °C/min, subsequent increase to 300 °C with 7 °C/min heating ramp, and then held for 10 min. The injector temperature was 250 °C. The carrier gas was helium, at a flow rate of 2.25 mL/min. The sample (1 μL) was injected without splitting. The MS was operated in electron impact ionization mode (70 eV). The mass ranged from m/z 50 amu to m/z 500 amu. The solvent delay time was 4 min. Most components, such as the sesquiterpenoids, were identified by comparing their mass spectra with the NIST2017 library data for the GC-MS system. 2-(2-phenylethyl)chromones were identified with MS/MS fragmentation and based on the accurate mass and mass spectrometric fragmentation patterns combined with the relevant literature data [34]. This was performed by calculating the peak area of each compound, which was then used to calculate its relative content.

Total chromone content

According to the method of Ma et al. [11], 2-(2-phenylethyl)chromone was weighed in appropriate amounts and prepared in methanol as standard solutions at concentrations of 1~6 mg/L and the methanol reagent was used as a blank control. The prepared standard solution was used to determine the absorbance at 250 nm using UV-Vis (GE, Ultrospec 2100 PRO, Amersharm Bioscience, NJ, USA) with X (mg/L) concentration as the horizontal coordinate and absorbance Y as the vertical coordinate to construct the standard curve. Subsequently, 0.2 g of IA and WA were added to 25 mL of ethanol, respectively, heated and extracted at reflux for 2 h, filtered, concentrated, and repeated 3 times to give alcohol-soluble extracts, dissolved in methanol, and the amount of total chromone in each was determined by measuring the absorbance. The experiments were repeated thrice and the results were expressed as mean ± standard deviation. The data were analyzed using the independent samples *t*-test of the SPSS software; if *P*< 0.05, there was a significant difference existing between the samples, otherwise, there was no significant difference.

#### 4.2.2. Bioactivity Measurements

Preparation of alcohol-soluble extract

A total of 5 g of IA and WA was measured and 200 mL of ethanol added. The mixture was boiled for 60 min after waiting for 60 min, filtered, and the process repeated twice. Then, the filtrates were combined and concentrated under reduced pressure and dried for 24 h to obtain the agarwood extracts for the bioactivity measurements.

Determination of antioxidant capacity

For the DPPH and ABTS^+^ radical scavenging experiments, various quantities of alcohol-soluble extracts were utilized in accordance with the approach used by Silva et al. [35]. Meanwhile, different ascorbic acid concentrations were used as positive controls (Appendix B).

Determination of acetylcholinesterase activity inhibition

The method of Wang et al. [36] was used with modification. Following the preparation of the sample solutions using various concentrations from the alcohol-soluble extracts, an assay for the inhibition of anti-acetylcholine ester activity was conducted. Tacrine was utilized as a positive control (Appendix B).

Determination of anti-α-glucosidase activity inhibition

The method of Ting et al. was used, with modification [37], to test the various concentrations of alcohol-soluble extracts of IA and WA for anti-α-glucosidase activity. A positive control was performed with an acarbose solution of the same concentration (Appendix B).

Data processing

A triple experiment was conducted and the mean and standard deviation (SD) were calculated. In addition, GraphPad Prism software was utilized to compute the IC_50_ values and create the charts. Additionally, independent samples t-tests were run on the data using SPSS software, with *p* < 0.05 indicating a significant difference between the samples and *p* > 0.05 indicating no significant difference between the samples.

## 5. Conclusions

Agarwood induced by fungal inducers has a chemical composition similar to that of wild agarwood, with a high content of sesquiterpenes and chromones. Its antioxidant activity, anti-acetylcholinesterase activity, and anti-alpha-glucosidase activity are excellent and very similar to that of wild agarwood. Therefore, it can be used as a substitute for wild agarwood in the development of appropriate Chinese patent medicines for human health.

## Figures and Tables

**Figure 1 molecules-28-02922-f001:**
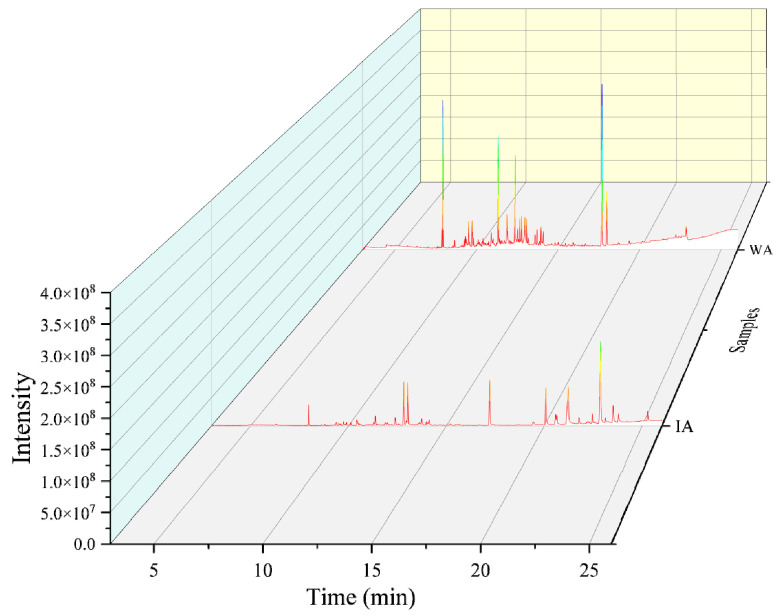
Total ion chromatogram (WA: wild agarwood; IA: induced agarwood).

**Figure 2 molecules-28-02922-f002:**
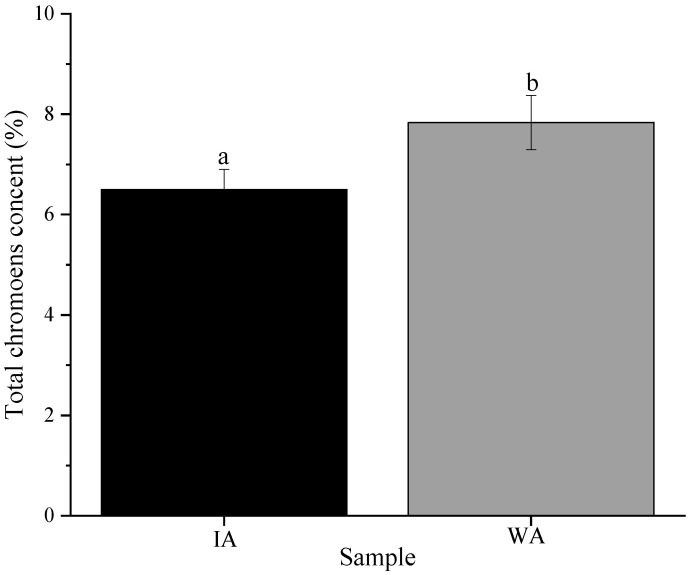
Total chromone content (WA: wild agarwood; IA: induced agarwood).

**Figure 3 molecules-28-02922-f003:**
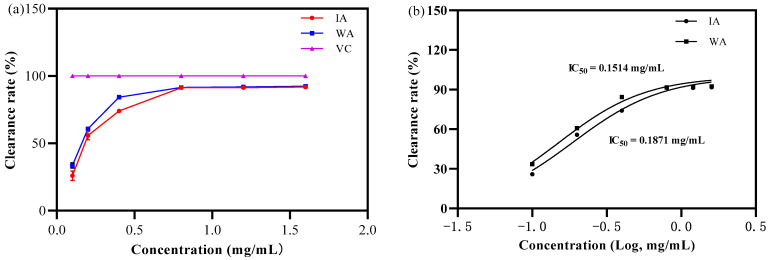
The clearance rate of DPPH free radicals (WA: wild agarwood; IA: induced agarwood; VC: ascorbic acid). (**a**) Represents that the sample’s DPPH removal rate increased with the mass concentration. (**b**) Represents the logarithmic value of the mass concentration and the calculated IC_50_ value of the DPPH.

**Figure 4 molecules-28-02922-f004:**
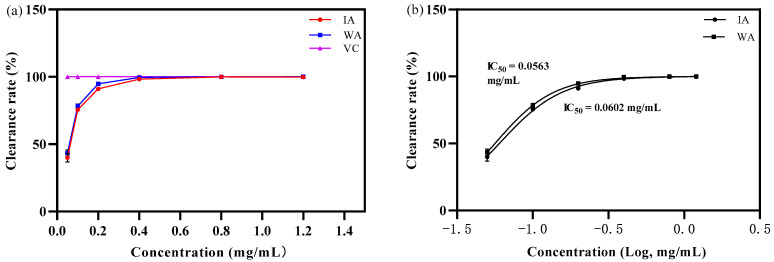
The clearance rate of ABTS^+^ free radicals (WA: wild agarwood; IA: induced agarwood; VC: ascorbic acid). (**a**) Represents that the sample ABTS^+^ removal rate increased with the mass concentration. (**b**) Represents the logarithmic value of the mass concentration and the calculated IC_50_ value of the ABTS^+^.

**Figure 5 molecules-28-02922-f005:**
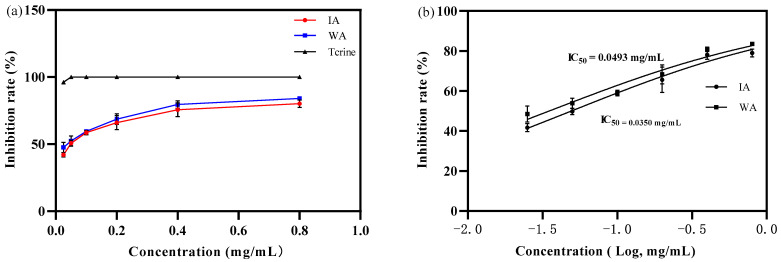
The inhibition rate of acetylcholinesterase (WA: wild agarwood; IA: induced agarwood). (**a**) Represents that the sample acetylcholinesterase inhibition rate increased with the mass concentration. (**b**) Represents the logarithmic value of the mass concentration and the calculated IC_50_ value of the acetylcholinesterase.

**Figure 6 molecules-28-02922-f006:**
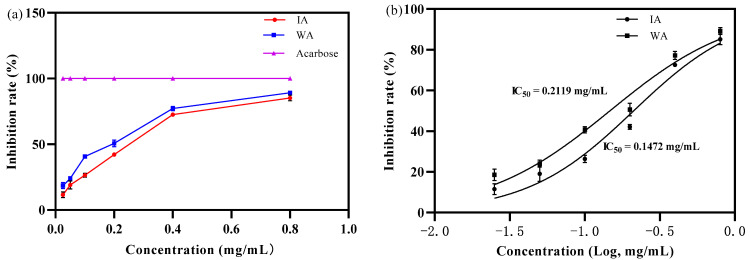
The inhibition rate of alpha-glucosidase (WA: wild agarwood; IA: induced agarwood). (**a**) Represents that the sample α-glucosidase inhibition rate increased with the mass concentration. (**b**) Represents the logarithmic value of mass concentration and the calculated IC_50_ value of the α-glucosidase.

**Figure 7 molecules-28-02922-f007:**
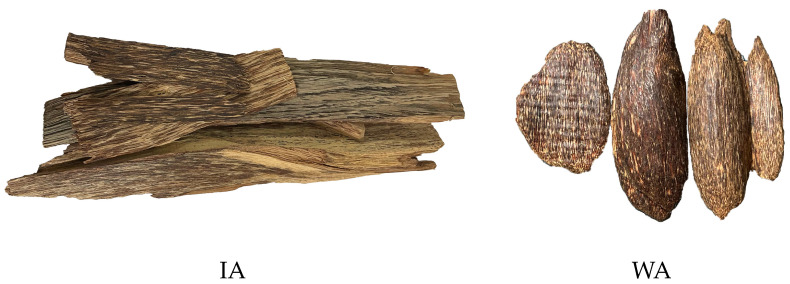
Induced agarwood (IA), wild agarwood (WA).

**Table 1 molecules-28-02922-t001:** Main chemical constituents.

IA			WA		
RT (min)	Component	Relative Content (%)	RT (min)	Component	Relative Content (%)
6.309	4-phenyl-2-Butanone	0.31	7.934	Phenol, 2,6-bis(1,1-dimethylethyl)-4-methyl-, methylcarbamate	8.11
9.518	Calarene epoxide	0.38	9.311	2-((2R,8R,8aS)-8,8a-Dimethyl-1,2,3,4,6,7,8,8a-octahydronaphthalen-2-yl)propan-2-ol	0.76
9.612	4-(1,5-dimethyl-1,4-hexadienyl)-1-methyl-Cyclohexene	0.30	9.354	Azulene, 1,2,3,5,6,7,8,8a-octahydro-1,4-dimethyl-7-(1-methylethenyl)-, [1S-(1.alpha.,7.alpha.,8a.beta.)]-	0.48
9.754	2-(2R,4aR,8aR)-4a,8-Dimethyl-1,2,3,4,4a,5,6,8a-octahydronaphthalen-2-yl)acrylaldehyde	0.76	9.363	Isolongifolen-5-one	0.51
9.845	3,4,4a,5,6,7-hexahydro-4a,5-dimethyl-3-(1-methylethenyl)-, [3S-(3.alpha.,4a.alpha.,5.alpha.)]-1(2H)-Naphthalenone	1.95	9.558	Spiro[4.5]dec-8-en-7-ol, 4,8-dimethyl-1-(1-methylethyl)-	0.28
9.902	Humulene epoxide I	0.55	9.724	Acorenone B	2.10
10.124	2-(4a,8-Dimethyl-1,2,3,4,4a,5,6,7-octahydro-naphthalen-2-yl)-prop-2-en-1-ol	0.50	10.098	Daucol	0.60
10.426	Ylangenal	1.18	10.163	Cyclohexanemethanol, 4-ethenyl-.alpha.,.alpha.,4-trimethyl-3-(1-methylethenyl)-, [1R-(1.alpha.,3.alpha.,4.beta.)]-	0.31
11.908	Arctiol	0.69	10.356	3-epi-Cedrenal	0.36
12.003	4,5,5a,6,6a,6b-hexahydro-4,4,6b-trimethyl-2-(1-methylethenyl)-2H-Cyclopropa[g]benzofuran	0.57	10.9	7-(2-Hydroxypropan-2-yl)-1,4a-dimethyldecahydronaphthalen-1-ol	1.25
12.392	Valerenic acid	1.64	11.006	2(3H)-Naphthalenone, 4,4a,5,6,7,8-hexahydro-4a,5-dimethyl-3-(1-methylethylidene)-, (4ar-cis)-	0.64
12.567	4-(3,3-Dimethyl-but-1-ynyl)-4-hydroxy-3,5,5-trimethyl-cyclohex-2-enone	0.29	11.237	(2S,6R)-2,6-Dimethyl-2-(2-(4-methylfuran-3-yl)ethyl)cyclohexanone	0.21
12.829	5,8-Dihydroxy-4a-methyl-4,4a,4b,5,6,7,8,8a,9,10-decahydro-2(3H)-phenanthrenone	18.56	11.325	2,4,6,7,8,8a-hexahydro-3,8-dimethyl-4-(1-methylethylidene)-(8S-cis)-5(1H)-azulenone	9.90
13.74	6-(1-Hydroxymethylvinyl)-4,8a-dimethyl-3,5,6,7,8,8a-hexahydro-1H-naphthalen-2-one	1.64	11.405	1,8-dimethyl-8,9-epoxy-4-isopropyl-Spiro[4.5]decan-7-one	0.58
13.981	(2aS,3aR,5aS,9bR)-2a,5a,9-Trimethyl-2a,4,5,5a,6,7,8,9b-octahydro-2H-naphtho[1,2-b]oxireno[2,3-c]furan	0.82	11.545	3.beta.,9.beta.-Dihydroxy-3,5.alpha.,8-trimethyltricyclo[6.3.1.0(1,5)]dodecane	0.29
15.657	2-hydroxy-5-(3-methyl-2-butenyl)-4-(1-methylethenyl)-2,4,6-Cycloheptatrien-1-one	0.29	11.869	1,1,4,7-Tetramethyldecahydro-1H-cyclopropa[e]azulene-4,7-diol	4.27
17.216	2-(2-phenylethyl)chromone	9.46	12.755	5,8-Dihydroxy-4a-methyl-4,4a,4b,5,6,7,8,8a,9,10-decahydro-2(3H)-phenanthrenone	6.09
19.43	7-Methoxy-3-(p-methoxyphenyl)chromone	3.06	13.079	(4aS,7R)-7-(2-Hydroxypropan-2-yl)-1,4a-dimethyl-4,4a,5,6,7,8-hexahydronaphthalen-2(3H)-one	2.52
20.09	8-Methoxy-2-(2-phenylethyl)chromone	8.39	13.59	6-(1-Hydroxymethylvinyl)-4,8a-dimethyl-3,5,6,7,8,8a-hexahydro-1H-naphthalen-2-one	1.40
22.261	Supraene	0.35	13.705	Valerenic acid	1.64
22.469	6-Methoxy-2-(4-methoxyphenethyl)chromone	2.15	13.952	(2aS,3aR,5aS,9bR)-2a,5a,9-Trimethyl-2a,4,5,5a,6,7,8,9b-octahydro-2H-naphtho[1,2-b]oxireno[2,3-c]furan	2.09
22.886	6,7-Dimethoxy-2-(2-phenethyl)chromone	23.17	14.085	2,4,6-Cycloheptatrien-1-one, 2-hydroxy-5-(3-methyl-2-butenyl)-4-(1-methylethenyl)-	1.32
25.286	6,7-Dimethoxy-2-(4-methoxyphenethyl)chromone	2.53	17.993	2,2′-methylenebis[6-(1,1-dimethylethyl)-4-methyl-phenol	4.54
			22.197	Supraene	0.27
			22.84	6,7-Dimethoxy-2-(2-phenethyl)chromone	1.16

## Data Availability

Data is unavailable due to privacy or ethical restrictions.

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
