# Peer review of "How Closely Does Induced Agarwood’s Biological Activity Resemble That of Wild Agarwood?"

_molecules, 2023, doi:10.3390/molecules28072922_

Round 1
Reviewer 1 Report
This work described the volatile components, total chromone content, and the differences in the overall activities of wild agarwood and agarwood induced. However, some revisions should be well made:
Figures: (1) the figure notes should contain the letter presented in your figure. For example, in “Figure 3. The clearance rate of DPPH free radicals”, it’s better to provide additional information and differences between Figure 3 (a) and (b). Same below.
(2) In Figure 7, a picture of healthy uninoculated A. sinensis plants should be also provided.
Discussion: in this section, too many specific numbers are described, which are supposed to be contained in the "Results" section.
Line 79: “A. sinensis” should be unified with the “agarwood” above.
In Table S1 of supplementary material, the formula of compounds identified in WA was all missing, please complete it.
Line 80: “36 compounds were detected identified,” should be corrected to “36 compounds were detected and tentatively identified”
Line 330: The identification strategy of the volatiles should be clearly described. Were some of the volatiles identified by comparison with the spectra of chemical standards? Or were all volatile just tentatively identified by comparison of their mass spectra with the NIST2014 and Wiley 275 library data? The necessary information with identification levels should be provided by the authors.
Line 315: More information on IA, WA, and HA should be provided, including the habitat, the way of collection, and preservation. Regarding the pretreatment method, the authors should provide detailed information about their sample preparation steps rather than simply referencing to a previous study.
Author Response
Response to Reviewers’ Comments
This work described the volatile components, total chromone content, and the differences in the overall activities of wild agarwood and agarwood induced. However, some revisions should be well made:
Question 1: Figures: (1) the figure notes should contain the letter presented in your figure. For example, in “Figure 3. The clearance rate of DPPH free radicals”, it’s better to provide additional information and differences between Figure 3 (a) and (b). Same below.
Response 1: Thank you for your opinion. We have revised them in lines 138-141, 164-167, 183-186, and 201-204.
Question 2: (2) In Figure 7, a picture of healthy uninoculated A. sinensis plants should be also provided.
Response 2: Thank you for your suggestion. We have provided HA photos on line 134 and 473.
Question 3: Discussion: in this section, too many specific numbers are described, which are supposed to be contained in the "Results" section.
Response 3: Thank you for your suggestion. We have revised them in line 210-211, 242-243, 247-248, 269-270, 273-274, 293, and 296.
Question 4: Line 79: “A. sinensis” should be unified with the “agarwood” above.
Response 4: Thank you for your suggestion. Agarwood is a product of stress, in contrast to A. sinensis trees which can’t be named agarwood.
Question 5: In Table S1 of supplementary material, the formula of compounds identified in WA was all missing, please complete it.
Response 5: Thank you for your opinion. We have completed it.
Question 6: Line 80: “36 compounds were detected identified,” should be corrected to “36 compounds were detected and tentatively identified”
Response 6: Thank you for your opinion. We have completed the revision in the of 75-76.
Question 7: Line 330: The identification strategy of the volatiles should be clearly described. Were some of the volatiles identified by comparison with the spectra of chemical standards? Or were all volatile just tentatively identified by comparison of their mass spectra with the NIST2014 and Wiley 275 library data? The necessary information with identification levels should be provided by the authors.
Response 7: Thank you for your suggestion. I'm sorry I didn't describe the process of clearing identification. First of all, I need to apologize. The database we refer to is NIST17, not NIST2014 and Wiley 275 library data, which has been corrected in line 338 of the manuscript. Due to the identified components, it is difficult to buy standard products on the market. Most of the data are combined with the NIST2017 database, as well as the mass spectrometry ion fragment (As shown in the following picture).
Question 8: Line 315: More information on IA, WA, and HA should be provided, including the habitat, the way of collection, and preservation. Regarding the pretreatment method, the authors should provide detailed information about their sample preparation steps rather than simply referencing a previous study.
Response 8: Thank you for your opinion. We have completed the revision in lines 306-318, and 320.

Reviewer 2 Report
Dear authors
I have problems with yours representation of the GCMS Data. The most assignments of the compounds HA sample are definitely wrong.
Here is a table of yours assignments and theirs reported Kovats indices in the NIST database for a HP-5MS Type column.
| 7,775 | Pentadecan | 1500 |
| 8,592 | Octadecanoic acid | 2180 |
| 9,561 | 2,6,11- trimethyl- Dodecane | 1275 |
| 10,643 | 3-methyl-5- propyl-Nonane | 1052 |
| 10,754 | 2,7,10- trimethyl- Dodecane | 1376 |
| 11,797 | Nonadecane | 1900 |
| 12,545 | n- Hexadecanoic acid | 1960 |
| 12,979 | Heneicosane | 2100 |
| 14,713 | Oleic acid | 2140 |
| 14,732 | 7-methylene-Bicyclo[4.1.0]heptane | |
| 14,939 | 1-Hexadecanol,acetate | 2010 |
| 17,480 | 3-Ethyl-2,6,10-trimethylun- decane | |
| 17,816 | Nonyl tetradecyl ether | |
| 22,259 | Hexadecane | 1600 |
This sequence of elution is not possible. Thus the assignment is wrong!
Therefor I have serious doubts on the assignments of the two other samples.
Additionally, comparing Table and Figure 1, I assume that the labelling was confused.
Author Response
Response to Reviewers’ Comments
Thank you for your correction. I'm sorry for this mistake. All the hints and remarks have been carefully checked and considered in our revised “manuscript”, and the modifications are highlighted in yellow color.
Question: I have problems with yours representation of the GCMS Data. The most assignments of the compounds HA sample are definitely wrong. Here is a table of yours assignments and theirs reported Kovats indices in the NIST database for a HP-5MS Type column. This sequence of elution is not possible. Thus the assignment is wrong!Therefore I have serious doubts on the assignments of the two other samples. Additionally, comparing Table and Figure 1, I assume that the labelling was confused.
Response 1: Thank you for your correction. I'm sorry for this mistake. All the hints and remarks have been carefully checked and considered in our revised “manuscript”, and the modifications are highlighted in yellow color.
First of all, I didn't find out that the column information. The column information is corrected as follows: HP-5MS is corrected as DB-5MS in line 331 (as shown in the following picture). And, then correct the total ion flow diagram of HA (the raw data is below), please forgive my low-level error. The occurrence order of substances is caused by the author's wrong order in excel, which has been corrected in the table 1(or the following table). Moreover, we also checked the data of the other two groups. Thank you again, dear reviewer, for pointing out our mistakes. I wish you a smooth career and a happy life.
The following is the original data of the total ion flow diagram of HA:
X(Minutes) Y(Counts)
3.001 206657.9219
3.005 88933.63281
3.009 0
3.014 0
3.018 0
3.022 0
3.026 0
3.03 0
3.035 660.0400391
3.039 2990.49292
3.043 8083.466797
3.047 19262.79883
3.051 43603.94922
3.056 85958.9375
3.06 176246.625
3.064 336542.125
3.068 593009.25
3.072 983188.25
3.076 1547813.875
3.081 2354063.25
3.085 3390908.5
3.089 4432250
3.093 5127503.5
3.097 5508275
3.102 5755928.5
3.106 6086180
3.11 6444458.5
3.114 6811039.5
3.118 7195816
3.123 7500926
3.127 7713226
3.131 7833721
3.135 8021196.5
3.139 8194330
3.144 8274417.5
3.148 8290049.5
3.152 8342687
3.156 8384927.5
3.16 8384550.5
3.165 8358454.5
3.169 8332070
3.173 8262094
3.177 8275622
3.181 8315488
3.185 8309566.5
3.19 8255266
3.194 8204305.5
3.198 8159085
3.202 8119422
3.206 8094043.5
3.211 8068249.5
3.215 8062030.5
3.219 7979390.5
3.223 7902696.5
3.227 7830068
3.232 7803385
3.236 7820015
3.24 7759826
3.244 7679570
3.248 7645047
3.253 7588590.5
3.257 7553536.5
3.261 7509493.5
3.265 7510305.5
3.269 7489162
3.274 7509635.5
3.278 7481039
3.282 7423484.5
3.286 7390951
3.29 7408223.5
3.294 7383813
3.299 7318054
3.303 7201505
3.307 7216249
3.311 7269987.5
3.315 7292906
3.32 7231184
3.324 7162696
3.328 7123509.5
3.332 7118702.5
3.336 7034147.5
3.341 6931619.5
3.345 6891893.5
3.349 6890460
3.353 6880704.5
3.357 6832164
3.362 6811624.5
3.366 6812542
3.37 6798639.5
3.374 6792237
3.378 6764949
3.383 6771570
3.387 6735390.5
3.391 6647277
3.395 6590260.5
3.399 6545715
3.403 6554434.5
3.408 6501511.5
3.412 6449578
3.416 6441867.5
3.42 6428924.5
3.424 6423287.5
3.429 6426805.5
3.433 6413048
3.437 6385739.5
3.441 6370323
3.445 6339915
3.45 6334700
3.454 6261073.5
3.458 6198997
3.462 6169847.5
3.466 6163616
3.471 6149221.5
3.475 6159727
3.479 6197135.5
3.483 6152331
3.487 6088740
3.492 6080809.5
3.496 6062586
3.5 5944749
3.504 5859468.5
3.508 5811940
3.512 5847307.5
3.517 5926863
3.521 5955663.5
3.525 5964299.5
3.529 5925394
3.533 5870214.5
3.538 5830825
3.542 5797223
3.546 5773458.5
3.55 5732961
3.554 5705470.5
3.559 5686802
3.563 5693054.5
3.567 5682678.5
3.571 5645421.5
3.575 5602298
3.58 5632065.5
3.584 5632925.5
3.588 5596236
3.592 5544209.5
3.596 5497677.5
3.601 5494113
3.605 5499136.5
3.609 5457888
3.613 5473067.5
3.617 5450132
3.622 5412463.5
3.626 5339091
3.63 5277817
3.634 5291670
3.638 5336661.5
3.642 5348562
3.647 5319284
3.651 5305633
3.655 5302976
3.659 5267597
3.663 5244120
3.668 5220255
3.672 5194063
3.676 5171609.5
3.68 5148275.5
3.684 5137363.5
3.689 5107620.5
3.693 5057824.5
3.697 5027392
3.701 5026401
3.705 5030773.5
3.71 5009575.5
3.714 4985019.5
3.718 4973744.5
3.722 4966185
3.726 4948571
3.731 4925623
3.735 4884462
3.739 4916208
3.743 4929558.5
3.747 4938841.5
3.751 4935041.5
3.756 4901002
3.76 4876377
3.764 4844262.5
3.768 4812663.5
3.772 4828047
3.777 4830596
3.781 4823434
3.785 4792320
3.789 4756215.5
3.793 4737864
3.798 4741368
3.802 4789625
3.806 4762353
3.81 4713661
3.814 4678530.5
3.819 4696902.5
3.823 4677917
3.827 4656601.5
3.831 4634927.5
3.835 4620621
3.84 4645856.5
3.844 4670058.5
3.848 4626012.5
3.852 4597483
3.856 4602063
3.86 4645797
3.865 4629230
3.869 4592919.5
3.873 4550864.5
3.877 4529462
3.881 4530431.5
3.886 4532071.5
3.89 4538433
3.894 4475144.5
3.898 4416856
3.902 4351572.5
3.907 4382364.5
3.911 4429230.5
3.915 4451132
3.919 4424327
3.923 4425550.5
3.928 4375588.5
3.932 4322115.5
3.936 4289377
3.94 4308972
3.944 4323326
3.949 4328121
3.953 4340138.5
3.957 4329884
3.961 4269537
3.965 4222618.5
3.969 4229510
3.974 4257340
3.978 4271926.5
3.982 4221822
3.986 4194034.25
3.99 4182541.25
3.995 4179209.75
3.999 4160530.75
4.003 4155985.75
4.007 4171298.5
4.011 4173151.5
4.016 4124785.25
4.02 4095410.25
4.024 4177540.5
4.028 4140378.25
4.032 4113018.5
4.037 4092671.75
4.041 4109743.25
4.045 4079757
4.049 4070857.75
4.053 4084046
4.058 4120603.75
4.062 4131780.25
4.066 4132188.25
4.07 4099624
4.074 4068573.25
4.078 4031307.25
4.083 4025696
4.087 3994903.75
4.091 3961266.25
4.095 3959700.25
4.099 3940588.75
4.104 3986446.5
4.108 3986637.25
4.112 3993000.75
4.116 3957765.75
4.12 3931746.5
4.125 3889364.75
4.129 3842321.75
4.133 3855459.5
4.137 3866097.75
4.141 3882789.25
4.146 3865902.5
4.15 3854094.25
4.154 3876595
4.158 3922416.5
4.162 3912061.25
4.167 3887546.5
4.171 3881802.25
4.175 3847711.75
4.179 3812200.75
4.183 3788860.75
4.187 3782235
4.192 3776422
4.196 3767252
4.2 3742596
4.204 3733716.75
4.208 3759802.5
4.213 3784838
4.217 3775933
4.221 3719479
4.225 3703153
4.229 3719915.5
4.234 3732586.75
4.238 3705844.75
4.242 3706236.5
4.246 3716723.5
4.25 3720249.75
4.255 3728546.25
4.259 3724883
4.263 3742639
4.267 3694951.25
4.271 3648083.25
4.276 3626487.25
4.28 3657325.5
4.284 3647187.75
4.288 3642465.75
4.292 3664566.5
4.296 3678723.75
4.301 3666220.25
4.305 3658007
4.309 3672909.75
4.313 3640430.75
4.317 3620176
4.322 3610163.5
4.326 3610060.25
4.33 3609910.75
4.334 3606571.75
4.338 3581536.75
4.343 3577824
4.347 3612142.25
4.351 3620542.25
4.355 3600945.5
4.359 3569903.5
4.364 3604936.25
4.368 3601521.25
4.372 3602084.25
4.376 3580847.75
4.38 3586056.75
4.385 3585704.5
4.389 3615539.25
4.393 3610258.75
4.397 3638981.25
4.401 3646759
4.405 3681842
4.41 3754236.75
4.414 3891775.75
4.418 4026231.25
4.422 4158365
4.426 4246100.5
4.431 4373498.5
4.435 4399473.5
4.439 4436416
4.443 4450785
4.447 4459027
4.452 4451635.5
4.456 4454486.5
4.46 4436505.5
4.464 4441634
4.468 4402647.5
4.473 4365342
4.477 4318646.5
4.481 4348596
4.485 4346623.5
4.489 4341934
4.494 4364403
4.498 4375247.5
4.502 4365370
4.506 4341864.5
4.51 4327021
4.515 4305266.5
4.519 4338813.5
4.523 4418468
4.527 4451602
4.531 4456242
4.535 4440143.5
4.54 4442973
4.544 4445841.5
4.548 4465087
4.552 4484885
4.556 4449628
4.561 4429164.5
4.565 4423943
4.569 4451113.5
4.573 4470547.5
4.577 4511084
4.582 4443008.5
4.586 4425141
4.59 4395437
4.594 4421437
4.598 4457536
4.603 4509302
4.607 4534138.5
4.611 4498393
4.615 4507470.5
4.619 4536538.5
4.624 4602166.5
4.628 4657474.5
4.632 4622013
4.636 4579631
4.64 4539671.5
4.644 4560938.5
4.649 4594718.5
4.653 4631227
4.657 4604359
4.661 4595620
4.665 4588444
4.67 4582707
4.674 4610667.5
4.678 4653043
4.682 4711709.5
4.686 4705504
4.691 4668748
4.695 4668513
4.699 4716654
4.703 4766770
4.707 4769050
4.712 4738789.5
4.716 4753574.5
4.72 4796310.5
4.724 4814471.5
4.728 4778373.5
4.733 4764665.5
4.737 4801287
4.741 4841834.5
4.745 4810824.5
4.749 4807169.5
4.753 4825766.5
4.758 4862168
4.762 4888283
4.766 4969733.5
4.77 4987471
4.774 4986041
4.779 4919428.5
4.783 4878065
4.787 4885509.5
4.791 4949202
4.795 4970536.5
4.8 4963925
4.804 4933627.5
4.808 4947462.5
4.812 4950650
4.816 5002356.5
4.821 5041051.5
4.825 5016241
4.829 4978673
4.833 5011028.5
4.837 5033417.5
4.842 5034104
4.846 5043178
4.85 5092610.5
4.854 5101156.5
4.858 5121362
4.862 5151665.5
4.867 5178351.5
4.871 5169203.5
4.875 5174331.5
4.879 5243745.5
4.883 5272003
4.888 5280512
4.892 5255537.5
4.896 5283123.5
4.9 5283398
4.904 5288319.5
4.909 5293101.5
4.913 5315128
4.917 5301157
4.921 5329550.5
4.925 5388325.5
4.93 5395254.5
4.934 5361727.5
4.938 5299081.5
4.942 5285255.5
4.946 5306975
4.951 5369454
4.955 5410562.5
4.959 5387504.5
4.963 5394252.5
4.967 5403562
4.971 5459444
4.976 5461724
4.98 5445308.5
4.984 5461845.5
4.988 5504974
4.992 5541922.5
4.997 5576203
5.001 5564462
5.005 5584346.5
5.009 5611572.5
5.013 5644579
5.018 5627686.5
5.022 5659177.5
5.026 5702369
5.03 5731013
5.034 5731080.5
5.039 5723392.5
5.043 5765626.5
5.047 5852932.5
5.051 5867187
5.055 5827552
5.06 5762697
5.064 5793893
5.068 5756535
5.072 5752439
5.076 5801660
5.081 5870034.5
5.085 5886285
5.089 5851178
5.093 5864217
5.097 5887634.5
5.101 5951907
5.106 5997236
5.11 6041087
5.114 6055479
5.118 6072233.5
5.122 6027125
5.127 5991754
5.131 5988530
5.135 5979738
5.139 5932751.5
5.143 5950756.5
5.148 5982181
5.152 5969565
5.156 5945838.5
5.16 5955367.5
5.164 5978074
5.169 6025237
5.173 6060286
5.177 6089059.5
5.181 6109047.5
5.185 6127314.5
5.19 6149273
5.194 6182071.5
5.198 6187434
5.202 6133486
5.206 6134424
5.21 6119913.5
5.215 6140615
5.219 6137632
5.223 6136985.5
5.227 6133705.5
5.231 6193387.5
5.236 6240144
5.24 6226603.5
5.244 6174624.5
5.248 6156813
5.252 6142519.5
5.257 6131897.5
5.261 6142933.5
5.265 6157285
5.269 6139429
5.273 6137466
5.278 6149038
5.282 6132654.5
5.286 6089451.5
5.29 6103847.5
5.294 6151527
5.299 6207240.5
5.303 6244896
5.307 6271849.5
5.311 6261339
5.315 6282716.5
5.319 6325230.5
5.324 6330002.5
5.328 6357363.5
5.332 6360380.5
5.336 6301454
5.34 6285371.5
5.345 6344752
5.349 6381711.5
5.353 6370673.5
5.357 6346811
5.361 6317629.5
5.366 6297662.5
5.37 6286304.5
5.374 6291704
5.378 6339155.5
5.382 6378766.5
5.387 6380118.5
5.391 6427730.5
5.395 6540149
5.399 6605357
5.403 6632597.5
5.408 6613795.5
5.412 6645937
5.416 6652636.5
5.42 6665639.5
5.424 6667800.5
5.428 6641481
5.433 6599088.5
5.437 6542451.5
5.441 6533479.5
5.445 6547971
5.449 6579254.5
5.454 6529483.5
5.458 6525886
5.462 6524415
5.466 6519031.5
5.47 6530922
5.475 6513000.5
5.479 6509104
5.483 6496220
5.487 6533734.5
5.491 6539565
5.496 6532643
5.5 6493366.5
5.504 6495003.5
5.508 6491973
5.512 6449066.5
5.517 6443951.5
5.521 6486680
5.525 6563787.5
5.529 6579582
5.533 6566650.5
5.538 6527792.5
5.542 6528075
5.546 6570410
5.55 6585369.5
5.554 6549643
5.558 6540955.5
5.563 6570716.5
5.567 6594238.5
5.571 6594578.5
5.575 6580836.5
5.579 6583114.5
5.584 6587710.5
5.588 6633252
5.592 6672600
5.596 6738649
5.6 6739861.5
5.605 6764928
5.609 6796526.5
5.613 6922411
5.617 7235926.5
5.621 7651983
5.626 8007374.5
5.63 8133976.5
5.634 8002059
5.638 7693158
5.642 7327717.5
5.647 7018056.5
5.651 6821493.5
5.655 6719977
5.659 6730128.5
5.663 6783517.5
5.667 6806667
5.672 6780310.5
5.676 6726883
5.68 6689180.5
5.684 6652632.5
5.688 6610278.5
5.693 6614184.5
5.697 6624864.5
5.701 6602433.5
5.705 6620067
5.709 6622903.5
5.714 6634516.5
5.718 6601508
5.722 6615919.5
5.726 6646478
5.73 6670469
5.735 6623832.5
5.739 6590295.5
5.743 6572455.5
5.747 6572542.5
5.751 6593171.5
5.756 6585818
5.76 6553856
5.764 6623895.5
5.768 6767219
5.772 7007841.5
5.776 7129379
5.781 7114629
5.785 6940894
5.789 6726816.5
5.793 6584805.5
5.797 6508694
5.802 6530101.5
5.806 6532448.5
5.81 6552035.5
5.814 6540156.5
5.818 6552648.5
5.823 6541648.5
5.827 6573154
5.831 6572522.5
5.835 6530978.5
5.839 6487582.5
5.844 6467384.5
5.848 6478455.5
5.852 6494134.5
5.856 6546956
5.86 6538993
5.865 6489514.5
5.869 6400989
5.873 6425308.5
5.877 6423720.5
5.881 6431561
5.885 6404655.5
5.89 6398229
5.894 6378994
5.898 6393376.5
5.902 6367384.5
5.906 6347832
5.911 6368720.5
5.915 6431338
5.919 6383457
5.923 6362889
5.927 6337335
5.932 6286179.5
5.936 6270215
5.94 6263685
5.944 6283927.5
5.948 6241056
5.953 6259539
5.957 6284913.5
5.961 6324594
5.965 6293311
5.969 6255108.5
5.974 6288612.5
5.978 6269730.5
5.982 6177986.5
5.986 6100422.5
5.99 6075494
5.995 6062803
5.999 6072701.5
6.003 6110664
6.007 6247246
6.011 6385992
6.015 6498890
6.02 6520407.5
6.024 6494306.5
6.028 6335610
6.032 6226048.5
6.036 6096937.5
6.041 6056484
6.045 6085698
6.049 6099377.5
6.053 6103406
6.057 6085378.5
6.062 6068102
6.066 6144570.5
6.07 6197414
6.074 6227867.5
6.078 6205776
6.083 6175790.5
6.087 6122152
6.091 6053774.5
6.095 6014768.5
6.099 5997636.5
6.104 5924642.5
6.108 5874508.5
6.112 5916180.5
6.116 5987014.5
6.12 6048919.5
6.124 6050130.5
6.129 6017091.5
6.133 5935336
6.137 5834774
6.141 5779610
6.145 5810927.5
6.15 5822558.5
6.154 5793127
6.158 5776604
6.162 5791138.5
6.166 5778067.5
6.171 5791030.5
6.175 5786115.5
6.179 5755412.5
6.183 5736086.5
6.187 5705191
6.192 5737088.5
6.196 5733311.5
6.2 5737057
6.204 5721646.5
6.208 5767232
6.213 5723025.5
6.217 5662944
6.221 5652194.5
6.225 5682705.5
6.229 5724345.5
6.233 5695730
6.238 5660755.5
6.242 5621181
6.246 5639500
6.25 5650065.5
6.254 5632925.5
6.259 5618533.5
6.263 5622146.5
6.267 5603906
6.271 5648972
6.275 5683395
6.28 5699782
6.284 5682718.5
6.288 5659452.5
6.292 5652482.5
6.296 5645135
6.301 5596990
6.305 5576354
6.309 5683112
6.313 5920107.5
6.317 6133542.5
6.322 6243535
6.326 6138769
6.33 5917277.5
6.334 5725464
6.338 5636654
6.342 5576438
6.347 5570038
6.351 5528882
6.355 5517765
6.359 5463179.5
6.363 5422333
6.368 5339568.5
6.372 5292501
6.376 5264848
6.38 5234771
6.384 5195008
6.389 5180597
6.393 5150171.5
6.397 5184625
6.401 5377095.5
6.405 5676315.5
6.41 5826723
6.414 5677640.5
6.418 5314004
6.422 5053498
6.426 4885001.5
6.431 4830796
6.435 4814204
6.439 4806246.5
6.443 4864553.5
6.447 4943960.5
6.451 5019735
6.456 5007139.5
6.46 4981806
6.464 4943026
6.468 4939831
6.472 4909314
6.477 4920459.5
6.481 4925358.5
6.485 4961366.5
6.489 4971667
6.493 4962030.5
6.498 4915764.5
6.502 4899519
6.506 4860639
6.51 4858465.5
6.514 4868900.5
6.519 4952367
6.523 5011474
6.527 4967971.5
6.531 4830996
6.535 4736506
6.54 4693942
6.544 4711269
6.548 4693775
6.552 4663520
6.556 4619774
6.561 4631024
6.565 4665746
6.569 4793272
6.573 4953278.5
6.577 5135955.5
6.581 5121083
6.586 4856268
6.59 4603412.5
6.594 4455043
6.598 4409562.5
6.602 4390458.5
6.607 4334425.5
6.611 4294759
6.615 4269935.5
6.619 4313105
6.623 4343874
6.628 4314546.5
6.632 4333980.5
6.636 4343248.5
6.64 4496740.5
6.644 4643110
6.649 4682489.5
6.653 4518595
6.657 4332943
6.661 4229722
6.665 4135396.5
6.67 4118110
6.674 4071786
6.678 4080716.75
6.682 4105943.75
6.686 4143264.25
6.69 4177892.5
6.695 4219089
6.699 4183155
6.703 4157648.25
6.707 4078657.25
6.711 4067688.75
6.716 4015726
6.72 3982486
6.724 3976521
6.728 3996289.75
6.732 3990173.5
6.737 4014584.25
6.741 3988963.5
6.745 3987108.75
6.749 4007681
6.753 3975233.75
6.758 3937644.25
6.762 3911984.75
6.766 3885453.25
6.77 3883502.5
6.774 3859918.75
6.779 3832521.75
6.783 3848343.25
6.787 3860714.25
6.791 3836891.75
6.795 3817726.5
6.799 3771812.25
6.804 3734357.5
6.808 3755453.5
6.812 3749290
6.816 3740368.25
6.82 3732192.75
6.825 3695864.5
6.829 3692429.75
6.833 3674311.5
6.837 3681152.5
6.841 3693275.75
6.846 3719707.75
6.85 3706466.5
6.854 3628783.75
6.858 3603332.75
6.862 3629711
6.867 3591349.25
6.871 3560215
6.875 3520786.5
6.879 3563044
6.883 3561000.5
6.888 3587807.25
6.892 3636572.5
6.896 3686590.25
6.9 3645214.25
6.904 3585197
6.908 3536009.75
6.913 3461780.75
6.917 3444459.75
6.921 3463057.5
6.925 3470645.25
6.929 3431451.5
6.934 3399478
6.938 3357382.5
6.942 3327238.5
6.946 3286675.25
6.95 3280747.5
6.955 3317496.75
6.959 3329163.25
6.963 3337852.75
6.967 3282652
6.971 3235333.5
6.976 3263316.75
6.98 3283473.5
6.984 3312557.25
6.988 3302136.25
6.992 3315506
6.997 3324477.5
7.001 3303730.5
7.005 3319994.5
7.009 3322353.75
7.013 3320898.75
7.017 3305105.75
7.022 3361026.25
7.026 3306855.5
7.03 3250034.75
7.034 3158361.75
7.038 3114270.5
7.043 3142098.25
7.047 3199145.5
7.051 3360993
7.055 3672514.25
7.059 4023866
7.064 4150138.5
7.068 3927495.25
7.072 3525867.5
7.076 3246034.5
7.08 3125369.5
7.085 3093303.75
7.089 3105683.5
7.093 3080126
7.097 3103532.5
7.101 3126594.25
7.106 3111864.25
7.11 3053765.5
7.114 3059207.5
7.118 3073695
7.122 3122814.5
7.126 3133212.5
7.131 3110184
7.135 3077977.75
7.139 3042195.5
7.143 3046292.5
7.147 3019212.25
7.152 3045285.75
7.156 3042003
7.16 3019236.25
7.164 2968961.5
7.168 2918751
7.173 2902696
7.177 2865203.5
7.181 2883064.5
7.185 2869760.25
7.189 2873516.25
7.194 2861750.75
7.198 2915329.5
7.202 2925141.5
7.206 2939511.5
7.21 2904465.75
7.215 2911818.75
7.219 2896241.25
7.223 2860667.75
7.227 2826461.75
7.231 2774556.75
7.235 2755936.5
7.24 2731749.75
7.244 2728658.5
7.248 2702187.25
7.252 2692320.5
7.256 2677981.75
7.261 2702874.75
7.265 2713666.5
7.269 2697524.75
7.273 2675974.75
7.277 2668036.25
7.282 2643586.5
7.286 2604672.75
7.29 2614237.75
7.294 2601257.25
7.298 2641304.25
7.303 2661226.5
7.307 2669181.5
7.311 2694432
7.315 2703030.25
7.319 2722458
7.324 2682796
7.328 2651083.75
7.332 2627030.75
7.336 2630832.25
7.34 2609193.25
7.344 2670676.5
7.349 2675411.25
7.353 2735712.25
7.357 2710615.75
7.361 2666777.5
7.365 2622429.25
7.37 2609050.5
7.374 2633912.5
7.378 2657144.5
7.382 2626833.25
7.386 2625928.75
7.391 2617097.75
7.395 2593811
7.399 2636944
7.403 2669350.5
7.407 2746670
7.412 2763536.75
7.416 2773711.75
7.42 2700630.5
7.424 2618949.25
7.428 2612774
7.433 2616840
7.437 2656531.25
7.441 2749082.25
7.445 2776390.5
7.449 2802920
7.453 2792365
7.458 2774922
7.462 2747031
7.466 2714788
7.47 2739883.75
7.474 2748358.25
7.479 2828566.25
7.483 2872563
7.487 2847490.5
7.491 2826123
7.495 2764950
7.5 2714735.5
7.504 2710023
7.508 2691039.25
7.512 2699609.75
7.516 2598284.25
7.521 2527600.5
7.525 2536545.25
7.529 2507496.5
7.533 2546678.75
7.537 2515408.25
7.542 2489450
7.546 2436409.5
7.55 2414955
7.554 2388060
7.558 2422749.5
7.562 2371931
7.567 2337369.25
7.571 2416969.75
7.575 2553739.75
7.579 2875268.25
7.583 3239280.5
7.588 3583567
7.592 3699430.25
7.596 3618632.25
7.6 3270315
7.604 2942112.25
7.609 2702870
7.613 2594084.5
7.617 2500196.5
7.621 2499133.5
7.625 2503049
7.63 2519857
7.634 2501118
7.638 2447288.5
7.642 2413270.5
7.646 2417320.5
7.651 2378263
7.655 2388702
7.659 2430725.75
7.663 2503003.75
7.667 2547542.75
7.671 2652275.5
7.676 2760522
7.68 2816763.75
7.684 2847143.75
7.688 2894590.75
7.692 2975883.5
7.697 2954768.75
7.701 2899118.75
7.705 2833397
7.709 2779597
7.713 2853815.25
7.718 3077731
7.722 3345416.75
7.726 3516936.5
7.73 3461295.75
7.734 3345691.75
7.739 3290288.5
7.743 3471795.25
7.747 3743684.5
7.751 3920624
7.755 3978461.25
7.76 3671870.5
7.764 3321608
7.768 3006937.25
7.772 2726092.5
7.776 2571089.5
7.78 2480397.5
7.785 2414186.5
7.789 2362516.25
7.793 2339117
7.797 2352283.5
7.801 2399206.25
7.806 2507842.5
7.81 2623892.75
7.814 2692434.5
7.818 2715167.25
7.822 2697442.25
7.827 2620628.75
7.831 2539097.25
7.835 2472461.25
7.839 2414812.5
7.843 2396096.75
7.848 2336734.75
7.852 2310806
7.856 2332054.5
7.86 2387135
7.864 2404298.25
7.869 2429073.75
7.873 2437341.75
7.877 2372775.75
7.881 2281174.5
7.885 2232403.75
7.889 2218135.5
7.894 2250774.25
7.898 2345556.75
7.902 2482468.25
7.906 2627340.5
7.91 2753461.75
7.915 2720699.25
7.919 2640401.5
7.923 2540257.5
7.927 2465757
7.931 2415276.75
7.936 2373541.75
7.94 2332452.75
7.944 2304280.25
7.948 2284470
7.952 2237866
7.957 2228797.25
7.961 2253888.25
7.965 2256169.25
7.969 2223940.5
7.973 2310787
7.978 2351386
7.982 2370521.25
7.986 2297971
7.99 2246063
7.994 2192828.5
7.998 2208792.75
8.003 2194994.75
8.007 2188803.75
8.011 2235050.75
8.015 2270004.75
8.019 2296692.75
8.024 2318226.75
8.028 2257602.75
8.032 2256405.75
8.036 2121293.25
8.04 2162709.5
8.045 2178445.25
8.049 2163720.25
8.053 2143422
8.057 2165809
8.061 2115366.75
8.066 2139823.25
8.07 2142579
8.074 2131048.75
8.078 2104075
8.082 2130044.25
8.087 2119918
8.091 2214636.75
8.095 2325749.25
8.099 2488774.25
8.103 2511371.25
8.107 2487216.75
8.112 2366542
8.116 2305441
8.12 2247400.25
8.124 2218709
8.128 2190308.25
8.133 2174494.25
8.137 2154616.75
8.141 2123240.25
8.145 2091632.125
8.149 2061787.5
8.154 2007847.125
8.158 2006817.875
8.162 2046551.875
8.166 2073940.875
8.17 2130385.5
8.175 2145894.5
8.179 2173544
8.183 2165702.75
8.187 2175763.5
8.191 2179891.25
8.196 2162093.25
8.2 2161379.5
8.204 2145451.5
8.208 2154746.75
8.212 2155027.25
8.216 2230313.75
8.221 2224685.75
8.225 2273092.5
8.229 2270127.5
8.233 2225258.5
8.237 2239320
8.242 2192698.25
8.246 2222866.75
8.25 2315343
8.254 2392764
8.258 2533214.25
8.263 2487929.25
8.267 2426452.75
8.271 2345318.5
8.275 2250404.5
8.279 2134959
8.284 2084847.25
8.288 2047443.25
8.292 2045213.125
8.296 2074600.875
8.3 2086567.625
8.305 2085222.5
8.309 2080422.25
8.313 2058190.75
8.317 2050572
8.321 2081713.875
8.325 2124265.25
8.33 2147728.25
8.334 2139961.75
8.338 2130327.75
8.342 2155883.5
8.346 2147489.25
8.351 2106277.5
8.355 2090254
8.359 2122779.5
8.363 2101689.75
8.367 2129319.25
8.372 2193218.75
8.376 2198598.5
8.38 2204684.25
8.384 2190823.25
8.388 2210404.5
8.393 2245274.25
8.397 2131035.75
8.401 2039670.875
8.405 1983843.375
8.409 1969892
8.414 1980741.125
8.418 2005936.375
8.422 1971829.25
8.426 1967335.25
8.43 1970372.75
8.434 1983562.625
8.439 1979375.125
8.443 1948091.5
8.447 1964668.25
8.451 2005913.625
8.455 2040970.25
8.46 2028853.5
8.464 2043077.875
8.468 2012805.125
8.472 2013758.375
8.476 2001151.625
8.481 1973744.125
8.485 1954930.375
8.489 1983893.5
8.493 2022624.25
8.497 2162224.25
8.502 2247810
8.506 2328915.25
8.51 2372765.75
8.514 2306742
8.518 2206444.25
8.522 2135486
8.527 2041656.875
8.531 2033892
8.535 2000480.5
8.539 2036942.875
8.543 2021533.5
8.548 1948207.75
8.552 1901381.75
8.556 1930104.625
8.56 1906827.875
8.564 1987790.125
8.569 2030021.625
8.573 2160970.75
8.577 2407332.25
8.581 2725483
8.585 3081383.25
8.59 3264132.75
8.594 3246633.75
8.598 2994122.25
8.602 2612608.75
8.606 2326947
8.611 2083589.75
8.615 1984058.25
8.619 1965724.25
8.623 1954084.375
8.627 1934747.875
8.631 1879751
8.636 1894124.125
8.64 1952434
8.644 1996677.5
8.648 2041318
8.652 2083230
8.657 2106370.75
8.661 2119096
8.665 2221588
8.669 2218707.75
8.673 2228918
8.678 2207579.75
8.682 2174352.5
8.686 2157300
8.69 2149957.25
8.694 2093182.125
8.699 2109645.75
8.703 2115144.25
8.707 2113288.5
8.711 2077155.625
8.715 2119350.25
8.72 2193666
8.724 2188097.5
8.728 2158463
8.732 2154934.75
8.736 2142052.5
8.74 2097757.75
8.745 2133184
8.749 2141132
8.753 2132810.5
8.757 2076885.75
8.761 2073661.875
8.766 2037491.375
8.77 2022994.75
8.774 2038242.875
8.778 2052180.5
8.782 2059681.5
8.787 2101621.25
8.791 2107948.5
8.795 2136469.5
8.799 2093727.5
8.803 2064980.5
8.808 2069448.5
8.812 2072940.125
8.816 2048852.375
8.82 2040715
8.824 2043494.5
8.829 2061671.5
8.833 2095003.875
8.837 2076253
8.841 2058542.25
8.845 2067862
8.849 2045175.25
8.854 1978334.375
8.858 1956819.25
8.862 1967217
8.866 1920836.875
8.87 1892086.75
8.875 1873130.625
8.879 1877165.75
8.883 1885324.875
8.887 1930291.125
8.891 1964139.75
8.896 1954079.75
8.9 1991355.5
8.904 2015397.875
8.908 2018885
8.912 2034197.625
8.917 2087005.75
8.921 2096514.875
8.925 2089131.375
8.929 2060690.125
8.933 2084031.75
8.938 2117527.75
8.942 2055861.875
8.946 2079059.625
8.95 2041015.625
8.954 2079792.125
8.958 2137811.25
8.963 2120892.25
8.967 2079324.25
8.971 2024501.5
8.975 2008843.375
8.979 1999724.5
8.984 1961149.125
8.988 1970085.25
8.992 1936809
8.996 1923594
9 1918786
9.005 2004114.875
9.009 2030410.5
9.013 2061921.875
9.017 2054680.5
9.021 2057100.5
9.026 2045950.875
9.03 2045955
9.034 2099708.25
9.038 2216426.5
9.042 2293988
9.047 2414105.5
9.051 2493768
9.055 2427608.75
9.059 2371048
9.063 2295572.75
9.067 2210497.5
9.072 2162378.5
9.076 2135673.5
9.08 2134292.75
9.084 2111876
9.088 2105544.5
9.093 2111936
9.097 2153295.25
9.101 2215798.75
9.105 2298531.75
9.109 2307926.75
9.114 2283406.75
9.118 2203832.25
9.122 2119090.75
9.126 2048849.875
9.13 2003808
9.135 1992567.125
9.139 2000260
9.143 2008476.625
9.147 2006395.125
9.151 2012003.25
9.156 1978890.125
9.16 1983161.375
9.164 1975795.375
9.168 1952943
9.172 1986743.125
9.176 2095347.5
9.181 2189413.75
9.185 2251176.5
9.189 2349606
9.193 2429424
9.197 2464549.75
9.202 2411955.5
9.206 2419731.25
9.21 2405215.75
9.214 2337775.25
9.218 2246462.5
9.223 2139867.5
9.227 2111411.25
9.231 2027271.375
9.235 2011087
9.239 1974879.125
9.244 1997765.375
9.248 1994597.375
9.252 2008096.5
9.256 2049408
9.26 2064153.25
9.265 2089886.75
9.269 2117258.75
9.273 2093640.25
9.277 2065755.875
9.281 2049692.25
9.285 2021221.375
9.29 1945600.125
9.294 1971134.125
9.298 1995936.875
9.302 1964146.875
9.306 1991724.125
9.311 2007788.25
9.315 2027906.75
9.319 2026027.25
9.323 2081960.875
9.327 2141201.25
9.332 2228899.5
9.336 2358430.75
9.34 2404964.25
9.344 2485342.5
9.348 2592197.25
9.353 2540405.5
9.357 2524901.75
9.361 2488384.75
9.365 2492937
9.369 2418267.25
9.374 2353792.25
9.378 2293672.75
9.382 2260369.25
9.386 2203937.75
9.39 2190165.75
9.394 2093017.625
9.399 2108863.75
9.403 2146294.75
9.407 2149622.5
9.411 2200718.75
9.415 2200357
9.42 2222145.5
9.424 2214368.75
9.428 2230075.25
9.432 2236577.25
9.436 2242440.25
9.441 2159070.25
9.445 2120176.75
9.449 2109496.5
9.453 2108705
9.457 2114506
9.462 2172052.25
9.466 2156129.75
9.47 2148982.25
9.474 2104365.5
9.478 2063910
9.483 2063741.5
9.487 2087147.75
9.491 2036259.375
9.495 2033055.125
9.499 2047935
9.503 2078158.625
9.508 2114274
9.512 2121444
9.516 2151080.75
9.52 2168934
9.524 2205589.75
9.529 2206275.5
9.533 2225945.25
9.537 2322782.25
9.541 2397518
9.545 2549894.75
9.55 2775518.25
9.554 3056043.25
9.558 3286301.5
9.562 3377919.5
9.566 3295892.25
9.571 3018925.5
9.575 2752338.75
9.579 2445140.75
9.583 2300301
9.587 2235671
9.591 2190176.5
9.596 2176617.5
9.6 2210669.75
9.604 2219184.25
9.608 2328294.25
9.612 2456062.5
9.617 2632598
9.621 2820062.5
9.625 2916954.25
9.629 2908237
9.633 2838322.25
9.638 2716472
9.642 2612032.25
9.646 2579087.5
9.65 2536744.75
9.654 2515880.75
9.659 2518098
9.663 2463966.5
9.667 2428328
9.671 2363653.5
9.675 2313458.5
9.68 2278619
9.684 2274594.25
9.688 2344471.75
9.692 2410903.25
9.696 2472905.5
9.7 2525080.75
9.705 2562595.5
9.709 2582333.25
9.713 2579342
9.717 2547383.75
9.721 2421877.75
9.726 2284885.5
9.73 2161865.5
9.734 2158372.5
9.738 2131459.75
9.742 2144569.75
9.747 2185900
9.751 2260885.75
9.755 2357238
9.759 2368498
9.763 2375051.75
9.768 2368745.5
9.772 2292305
9.776 2193027.5
9.78 2167561.25
9.784 2134865.25
9.789 2175992.25
9.793 2192813.5
9.797 2213079.5
9.801 2241453.5
9.805 2248052.5
9.809 2223698.75
9.814 2183998.5
9.818 2197572.75
9.822 2171799.75
9.826 2147721.5
9.83 2091674.625
9.835 2079514.5
9.839 2110670.75
9.843 2122686
9.847 2134410.75
9.851 2128784
9.856 2196603.25
9.86 2207335.75
9.864 2224103
9.868 2243158.75
9.872 2219696.75
9.877 2199974.25
9.881 2230490.25
9.885 2224792.25
9.889 2143682
9.893 2130406.75
9.898 2161690.25
9.902 2164913.75
9.906 2179428.75
9.91 2141408.5
9.914 2140924.75
9.918 2181089.25
9.923 2206930.25
9.927 2200859.75
9.931 2219692.25
9.935 2272406.25
9.939 2354039.5
9.944 2330037.25
9.948 2323439
9.952 2302094.75
9.956 2284914.75
9.96 2262062.25
9.965 2238518.25
9.969 2244369.5
9.973 2249179.5
9.977 2177958
9.981 2116383.5
9.986 2087965
9.99 2081141.375
9.994 2064971.25
9.998 2141233
10.002 2176514.25
10.007 2202776.75
10.011 2286000.5
10.015 2357939.5
10.019 2415314
10.023 2453406.75
10.027 2468657.75
10.032 2494335.75
10.036 2428219
10.04 2406620.5
10.044 2435328.25
10.048 2412982.25
10.053 2384405.5
10.057 2368483.25
10.061 2328067.5
10.065 2281645
10.069 2275671.25
10.074 2252916.5
10.078 2256504.75
10.082 2167044.25
10.086 2093767.125
10.09 2126785
10.095 2168254.5
10.099 2231529.5
10.103 2284842.75
10.107 2334541.75
10.111 2336847
10.116 2381296.5
10.12 2377564
10.124 2307264.25
10.128 2265619.5
10.132 2251039.75
10.136 2192682
10.141 2187022
10.145 2192939.5
10.149 2265816.5
10.153 2376563.25
10.157 2478738.5
10.162 2584750.5
10.166 2750852.5
10.17 2851304
10.174 2872754.5
10.178 2887201.75
10.183 2737784
10.187 2646901
10.191 2557405.25
10.195 2438887.75
10.199 2379404.75
10.204 2355028.25
10.208 2381540.75
10.212 2372006.5
10.216 2411076.75
10.22 2476977.75
10.225 2586709
10.229 2671152.75
10.233 2696433
10.237 2623169.5
10.241 2627314.5
10.245 2631619.25
10.25 2581767.25
10.254 2453949.75
10.258 2397263.5
10.262 2333786.75
10.266 2317001.75
10.271 2353795.5
10.275 2301836
10.279 2289036.75
10.283 2364363.5
10.287 2400104.75
10.292 2392780.5
10.296 2377002.75
10.3 2358356.25
10.304 2376846.5
10.308 2345090
10.313 2389823.5
10.317 2435070
10.321 2461042.25
10.325 2527162.5
10.329 2536012.5
10.334 2474362.25
10.338 2420757
10.342 2315728.5
10.346 2278154.75
10.35 2283641
10.354 2335009.75
10.359 2359369.25
10.363 2385419.75
10.367 2441603
10.371 2519980.25
10.375 2538676.75
10.38 2601842.25
10.384 2544415.5
10.388 2479817.75
10.392 2415885.5
10.396 2356178.75
10.401 2328153
10.405 2350133
10.409 2426949
10.413 2426610.25
10.417 2463600.75
10.422 2520668.75
10.426 2583387
10.43 2560141.25
10.434 2546278.75
10.438 2540065
10.443 2490676
10.447 2419054
10.451 2466936.5
10.455 2467885.25
10.459 2448288.25
10.463 2492208.75
10.468 2497260
10.472 2500541.5
10.476 2462134.25
10.48 2471448.25
10.484 2416740.25
10.489 2406109.5
10.493 2400659.5
10.497 2410419.75
10.501 2447113
10.505 2435766.5
10.51 2462399.75
10.514 2407027.25
10.518 2375301.5
10.522 2334857
10.526 2288834.5
10.531 2297859.5
10.535 2340476.25
10.539 2334102
10.543 2337489.75
10.547 2363373
10.552 2367012.25
10.556 2404804.25
10.56 2441013.25
10.564 2431627
10.568 2419386.75
10.572 2417045.5
10.577 2443207
10.581 2378068.25
10.585 2345151.25
10.589 2363240.25
10.593 2351798.5
10.598 2378289
10.602 2340882.5
10.606 2368554.75
10.61 2450944
10.614 2523741.75
10.619 2611460.5
10.623 2864201.5
10.627 3244067.75
10.631 3656125
10.635 4055404.25
10.64 4346672
10.644 4395741
10.648 4182926.5
10.652 3842200.75
10.656 3404658.75
10.66 3049840.25
10.665 2705777
10.669 2528398.25
10.673 2538179.75
10.677 2445505.25
10.681 2443246.75
10.686 2457237.75
10.69 2475123.5
10.694 2465087
10.698 2501019.5
10.702 2455091.5
10.707 2499771.75
10.711 2504665.5
10.715 2553350.75
10.719 2604747.75
10.723 2644068.25
10.728 2733209.5
10.732 2806076.5
10.736 2907627.5
10.74 3063559.75
10.744 3300269.5
10.749 3532512.75
10.753 3632204.5
10.757 3666270.25
10.761 3550109.75
10.765 3465596.75
10.769 3257650.5
10.774 3078772.75
10.778 2858949.75
10.782 2746545.5
10.786 2693061.5
10.79 2645181.25
10.795 2553918.5
10.799 2481100
10.803 2420114
10.807 2349039.75
10.811 2361661.75
10.816 2323783.75
10.82 2322313.25
10.824 2359075
10.828 2342145.25
10.832 2386534.75
10.837 2425113.5
10.841 2396203
10.845 2364614.5
10.849 2325039.75
10.853 2290460
10.858 2312495
10.862 2300595.5
10.866 2349044.25
10.87 2362882.25
10.874 2437148.75
10.878 2481337
10.883 2486270.25
10.887 2503371
10.891 2504154
10.895 2545864.25
10.899 2584178
10.904 2576473.25
10.908 2564783.5
10.912 2545839.5
10.916 2544206.5
10.92 2578451.25
10.925 2666818.75
10.929 2841081
10.933 2900212.75
10.937 2995215.75
10.941 3020709.5
10.946 3049184.75
10.95 2942817.75
10.954 2802270.5
10.958 2714900.25
10.962 2666741.75
10.967 2570930.25
10.971 2519843.75
10.975 2466074
10.979 2458057.5
10.983 2459153
10.987 2356569.75
10.992 2330867.75
10.996 2301061
11 2323031
11.004 2321198.5
11.008 2404044
11.013 2407213.25
11.017 2442621.25
11.021 2452107.75
11.025 2472250.5
11.029 2473318.25
11.034 2557652.5
11.038 2605860.75
11.042 2650306.25
11.046 2652350.75
11.05 2586910.5
11.055 2518467.75
11.059 2467350.75
11.063 2431397.75
11.067 2366315.25
11.071 2359747.75
11.076 2358776.25
11.08 2322370.25
11.084 2332535
11.088 2421140
11.092 2475692
11.096 2517497.5
11.101 2551191.25
11.105 2608384
11.109 2687936.75
11.113 2756893.5
11.117 2811661.75
11.122 2809340
11.126 2772252.75
11.13 2818420.25
11.134 2808146.5
11.138 2822978.75
11.143 2772823.25
11.147 2777403.25
11.151 2791059
11.155 2851515.25
11.159 2916653
11.164 2920373.25
11.168 2969464.75
11.172 3063286
11.176 3043297.5
11.18 2962184.75
11.185 2938899.75
11.189 2827374
11.193 2717891
11.197 2652165.5
11.201 2653094.75
11.205 2638999.5
11.21 2618505.25
11.214 2591092.25
11.218 2648725.25
11.222 2631883
11.226 2626788.5
11.231 2620665.75
11.235 2658819.75
11.239 2632263.25
11.243 2620450.75
11.247 2589173.5
11.252 2666527
11.256 2655700.5
11.26 2688623.25
11.264 2721391.25
11.268 2730095.5
11.273 2601065.75
11.277 2539711.75
11.281 2523452.75
11.285 2505863.25
11.289 2581290.75
11.294 2650221.75
11.298 2725123.75
11.302 2836660
11.306 2908348.25
11.31 2978396.75
11.314 3033468.75
11.319 2994822.25
11.323 2969144.5
11.327 2999763.25
11.331 2898747.5
11.335 2861304
11.34 2844908.5
11.344 2736649.75
11.348 2779175.5
11.352 2797630.75
11.356 2770598
11.361 2817785.75
11.365 2868308.5
11.369 2860605.25
11.373 2886035.25
11.377 2903516.25
11.382 3004520.75
11.386 3135406.5
11.39 3220994.5
11.394 3284660.25
11.398 3344568.75
11.403 3374520.5
11.407 3284506
11.411 3113428.75
11.415 2884349.75
11.419 2743871.75
11.423 2673053
11.428 2665383.25
11.432 2644763.75
11.436 2672265.25
11.44 2701123.75
11.444 2746018.5
11.449 2772700
11.453 2866786.5
11.457 2874440.25
11.461 2897338
11.465 2750396.25
11.47 2738029.75
11.474 2692655.75
11.478 2690340.75
11.482 2646310.75
11.486 2653902
11.491 2660358.5
11.495 2709309.5
11.499 2705392.25
11.503 2695024.75
11.507 2728293.5
11.512 2767805.5
11.516 2842686.75
11.52 3013843.5
11.524 3208944.5
11.528 3325813.5
11.532 3516881
11.537 3621419.75
11.541 3652160.25
11.545 3555908.25
11.549 3378639
11.553 3088607.5
11.558 2902383.5
11.562 2787565.25
11.566 2770692.5
11.57 2679320.5
11.574 2687220
11.579 2692609
11.583 2754599.5
11.587 2772478.5
11.591 2804147.5
11.595 2794371.5
11.6 2813855.75
11.604 2782241.5
11.608 2814248.75
11.612 2864595.25
11.616 2810567.5
11.621 2706500.25
11.625 2774432.5
11.629 2755800
11.633 2765018.25
11.637 2700924.25
11.641 2650456.75
11.646 2670514.5
11.65 2677076.75
11.654 2674618.25
11.658 2740700.75
11.662 2773224.5
11.667 2836950.5
11.671 2854702.5
11.675 2847322.25
11.679 2876817.75
11.683 2933866.25
11.688 2972801.5
11.692 2970372.5
11.696 3091392.5
11.7 3095971.75
11.704 3008904.5
11.709 2965600.75
11.713 2995136
11.717 2990132.5
11.721 2941125.75
11.725 2861279
11.73 2760558
11.734 2734153.25
11.738 2681960.5
11.742 2685025.5
11.746 2701385.5
11.75 2706539.5
11.755 2730845.5
11.759 2735146.25
11.763 2750995.5
11.767 2891416.5
11.771 3117815.25
11.776 3425611.25
11.78 3765694.25
11.784 4086334.25
11.788 4278400.5
11.792 4408243
11.797 4362169
11.801 4111670
11.805 3762104.25
11.809 3366985
11.813 3083106.25
11.818 2901083
11.822 2746035
11.826 2697873.75
11.83 2680572.5
11.834 2682910.5
11.839 2681607.75
11.843 2699680
11.847 2687217.75
11.851 2699077
11.855 2715415
11.859 2722817.5
11.864 2703757.25
11.868 2670719
11.872 2627061.75
11.876 2658451.25
11.88 2660426.5
11.885 2750883.75
11.889 2756481.5
11.893 2726991.25
11.897 2743807.5
11.901 2727508.5
11.906 2703086.25
11.91 2701465.5
11.914 2723587.75
11.918 2729005.25
11.922 2801385.5
11.927 2853219.25
11.931 2812290.5
11.935 2862447.5
11.939 2890435.5
11.943 2887550.25
11.948 2867588.75
11.952 2807359.25
11.956 2819701.5
11.96 2803825.25
11.964 2820569.5
11.968 2814045.5
11.973 2773690.25
11.977 2734955.75
11.981 2699922
11.985 2732579.75
11.989 2700899.75
11.994 2691869
11.998 2730937.25
12.002 2704954.5
12.006 2693438
12.01 2670802.75
12.015 2686214.75
12.019 2638043
12.023 2709534.75
12.027 2673997.5
12.031 2671390.75
12.036 2642252.75
12.04 2691779.5
12.044 2687288
12.048 2656331.5
12.052 2713033.25
12.057 2714631.25
12.061 2764728.25
12.065 2777367
12.069 2800920
12.073 2825887.25
12.077 2876769.5
12.082 2876880.75
12.086 2928035
12.09 3018818
12.094 3108538
12.098 3112782.75
12.103 3109441.5
12.107 3256016
12.111 3344951
12.115 3416469.5
12.119 3425039.5
12.124 3397256.25
12.128 3350152.75
12.132 3314907
12.136 3203604
12.14 3112183
12.145 3037248
12.149 2964497
12.153 2929326
12.157 2927557
12.161 2875715.75
12.166 2870208.75
12.17 2904107.75
12.174 2992139.75
12.178 2994489.5
12.182 3037031.75
12.186 3119737.25
12.191 3149787.5
12.195 3149756.5
12.199 3143390.25
12.203 3065255.75
12.207 2999810
12.212 2986674.75
12.216 2890678.75
12.22 2908722.25
12.224 2866262.75
12.228 2831090.75
12.233 2771214.5
12.237 2796852.5
12.241 2821645
12.245 2814577
12.249 2872808.75
12.254 2887797
12.258 2949906.5
12.262 2992955.25
12.266 3075443.25
12.27 3148311.75
12.274 3133427.25
12.279 3100410.25
12.283 3155029.75
12.287 3185588
12.291 3185779.75
12.295 3210068.5
12.3 3200159
12.304 3262393
12.308 3221219.75
12.312 3149854.5
12.316 3103773.5
12.321 3039961.5
12.325 3053025
12.329 3074438.5
12.333 3139097.75
12.337 3190581.25
12.342 3217452
12.346 3224831
12.35 3184642.5
12.354 3163836
12.358 3162659
12.363 3053794.5
12.367 3034698
12.371 3038384
12.375 3027117.5
12.379 2994159
12.383 3005156.5
12.388 3052184.5
12.392 2997568.25
12.396 3047023
12.4 3078582.5
12.404 3073778
12.409 3105701
12.413 3077510
12.417 3038163
12.421 3027263.5
12.425 3027449.5
12.43 3040042.5
12.434 2990269
12.438 2967418
12.442 2953090.5
12.446 2970135
12.451 2940137.5
12.455 2922442
12.459 2944073
12.463 2954507.25
12.467 2987537
12.472 3111888
12.476 3216249.75
12.48 3396998
12.484 3764386.25
12.488 4261190.5
12.492 5119999.5
12.497 6330961
12.501 7976253
12.505 9909438
12.509 12402404
12.513 15526579
12.518 18983716
12.522 22948598
12.526 26899550
12.53 30792922
12.534 34254252
12.539 37330708
12.543 39309772
12.547 38855808
12.551 34918416
12.555 27524628
12.56 19123214
12.564 13048030
12.568 9660674
12.572 7920912
12.576 7098601
12.581 6448250
12.585 6062928.5
12.589 5790147.5
12.593 5599383
12.597 5376003.5
12.602 5236375.5
12.606 5049695.5
12.61 4893421.5
12.614 4677214.5
12.618 4656669.5
12.622 4630785
12.627 4676881.5
12.631 4657014.5
12.635 4702626
12.639 4846695.5
12.643 4979372.5
12.648 4914927
12.652 4903169.5
12.656 4677567.5
12.66 4414690
12.664 4302110.5
12.669 4161038.25
12.673 4093623.25
12.677 3884905.5
12.681 3847241.5
12.685 3752468
12.69 3778921.75
12.694 3765319.25
12.698 3705403.25
12.702 3656985
12.706 3559053.5
12.711 3604577.25
12.715 3634748.25
12.719 3583974.75
12.723 3640614.25
12.727 3603280
12.731 3534811.25
12.736 3561327.5
12.74 3535742.5
12.744 3505015.75
12.748 3578458
12.752 3701784
12.757 3720577.5
12.761 3737603.25
12.765 3811561.5
12.769 3837315.25
12.773 3862779
12.778 3817979.25
12.782 3756947.5
12.786 3772962.75
12.79 3788117
12.794 3790030.75
12.799 3691442.75
12.803 3731740
12.807 3719170.75
12.811 3710908.75
12.815 3708932.5
12.82 3612923.25
12.824 3495687.25
12.828 3457965.5
12.832 3365181.25
12.836 3357973.75
12.84 3356652.75
12.845 3295131.5
12.849 3270994.5
12.853 3289883.25
12.857 3310459.25
12.861 3211307.25
12.866 3158425.5
12.87 3128152.25
12.874 3070663.5
12.878 3163751.5
12.882 3164215.5
12.887 3152055.75
12.891 3222506.5
12.895 3300630.75
12.899 3362765.75
12.903 3354295.25
12.908 3319178.5
12.912 3340729
12.916 3385979.5
12.92 3364809
12.924 3374803
12.929 3378749.5
12.933 3325012.25
12.937 3244286.5
12.941 3248942.25
12.945 3255692.75
12.949 3311320
12.954 3420479
12.958 3637101.25
12.962 3862604.75
12.966 4162627
12.97 4636763
12.975 5048159
12.979 5362956.5
12.983 5392762
12.987 5156761.5
12.991 4808982
12.996 4468956
13 4157309.5
13.004 3891746
13.008 3712866.5
13.012 3549719
13.017 3532712.5
13.021 3437509
13.025 3311776.5
13.029 3311257.5
13.033 3225750.75
13.038 3174403
13.042 3076114.5
13.046 3103887.75
13.05 3080680.25
13.054 3054830.5
13.058 3087021.25
13.063 3075275.5
13.067 3162013.25
13.071 3252326
13.075 3255178.75
13.079 3291933.25
13.084 3311304.75
13.088 3299473
13.092 3236397.75
13.096 3244431.75
13.1 3239180.5
13.105 3185759.5
13.109 3166998.75
13.113 3171934.25
13.117 3119111.75
13.121 3021780
13.126 2967482
13.13 2980050
13.134 2888039.5
13.138 2889165.25
13.142 2830649.5
13.147 2825606.75
13.151 2923388.25
13.155 2908298.5
13.159 2905311.5
13.163 2930412.75
13.167 2906744.25
13.172 2931933.25
13.176 2919338
13.18 2881909.75
13.184 2890544.25
13.188 2944339.75
13.193 2941990.5
13.197 2967844.75
13.201 2955026.5
13.205 2923458
13.209 2931495
13.214 2902185.25
13.218 2962793
13.222 2972253.5
13.226 2978236.25
13.23 3017328.75
13.235 3045984.75
13.239 3042687
13.243 3068018.25
13.247 3054795
13.251 2968337.75
13.256 2927686.25
13.26 2894872.5
13.264 2836473.25
13.268 2811295.25
13.272 2802897
13.276 2823928.75
13.281 2792637.75
13.285 2833695
13.289 2822605.25
13.293 2876051
13.297 2907439
13.302 2930871.25
13.306 2920806
13.31 2921394.5
13.314 2863853.25
13.318 2869892.75
13.323 2928036.5
13.327 2819829.25
13.331 2840950.75
13.335 2781170.75
13.339 2713738.75
13.344 2691767.75
13.348 2715116.75
13.352 2768297
13.356 2779730.75
13.36 2905986.75
13.365 2934935.75
13.369 2998626
13.373 3014171.75
13.377 3034025
13.381 3021714
13.385 2957478.5
13.39 2952390.5
13.394 3002612.5
13.398 2981667.75
13.402 2923275.25
13.406 2878923.25
13.411 2904441
13.415 2894672
13.419 2895094
13.423 2903913.25
13.427 2970496.5
13.432 3020638.25
13.436 3098972.75
13.44 3245387.5
13.444 3366795.75
13.448 3479429.5
13.453 3507367.5
13.457 3547513.5
13.461 3561722.25
13.465 3544976.5
13.469 3549927.25
13.474 3509380.75
13.478 3400018
13.482 3391322
13.486 3366561.25
13.49 3404578
13.494 3388041.25
13.499 3332541.25
13.503 3277553.25
13.507 3238596.75
13.511 3174830.25
13.515 3112557.75
13.52 3074204.25
13.524 2995462.25
13.528 2935271.5
13.532 2969545.5
13.536 2948081.25
13.541 2934892.5
13.545 2984210.25
13.549 2954168.5
13.553 2944651.25
13.557 2988064.5
13.562 3007270.75
13.566 2991864
13.57 3015443.5
13.574 2972732
13.578 2965409.5
13.583 2990572.5
13.587 2971926.5
13.591 2959126.5
13.595 2869761.25
13.599 2870617.25
13.603 2888044
13.608 2861888.5
13.612 2780459.25
13.616 2770291.75
13.62 2803766
13.624 2851901
13.629 2898133
13.633 2884192.75
13.637 2894817.25
13.641 2889867.5
13.645 2893933.75
13.65 2890912.5
13.654 2907118.25
13.658 3020663.25
13.662 3051093.75
13.666 3108999.25
13.671 3136055.5
13.675 3157704.5
13.679 3197210
13.683 3147331.5
13.687 3191691.5
13.692 3241744.25
13.696 3209439.25
13.7 3303142
13.704 3436667.5
13.708 3581985.75
13.712 3700624.75
13.717 3868989.25
13.721 3942405
13.725 3936494.25
13.729 3859121.75
13.733 3872739
13.738 3806262.5
13.742 3764770.75
13.746 3691338
13.75 3600459
13.754 3507926.5
13.759 3518993.25
13.763 3419558.5
13.767 3284049.75
13.771 3154769.75
13.775 3040455.25
13.78 3031959.5
13.784 2999388
13.788 2954331.25
13.792 2929174
13.796 2879352
13.801 2909023.5
13.805 2970238.5
13.809 2932782.5
13.813 2900630.75
13.817 2954614.75
13.821 3092912
13.826 3324321.5
13.83 3495971.25
13.834 3646066.75
13.838 3739169.25
13.842 3741678
13.847 3604620.75
13.851 3468788.25
13.855 3306230.25
13.859 3163505.5
13.863 3021082
13.868 2942547.5
13.872 2874874.5
13.876 2832958
13.88 2895428.5
13.884 2886772.25
13.889 2963300.5
13.893 3024486.5
13.897 3154104.25
13.901 3204106.5
13.905 3146627.75
13.91 3145756.25
13.914 3212842.5
13.918 3075749.25
13.922 3056583.25
13.926 2933265
13.93 2928538
13.935 2911508.5
13.939 2844871.75
13.943 2824456.5
13.947 2854915
13.951 2881700.5
13.956 2877190.25
13.96 2913002.5
13.964 2968607.25
13.968 2924085
13.972 2914101
13.977 2929463.75
13.981 2932466
13.985 2908382.75
13.989 2836615
13.993 2892940.25
13.998 2897526
14.002 2897326.5
14.006 2905465
14.01 2970347.25
14.014 3032763.75
14.019 3048545.25
14.023 3054109.25
14.027 3057261.5
14.031 2985219.75
14.035 2970109.75
14.039 2933193.25
14.044 2946729
14.048 2949686.25
14.052 2938770.75
14.056 2936071.25
14.06 2926297
14.065 2887482.5
14.069 2832544.5
14.073 2826292.5
14.077 2792525.75
14.081 2793345.5
14.086 2757227.75
14.09 2699718.5
14.094 2641588.25
14.098 2672413.5
14.102 2638698.75
14.107 2664508.25
14.111 2665069
14.115 2670616.75
14.119 2661814
14.123 2672078
14.128 2751891
14.132 2803536.5
14.136 2789607.5
14.14 2780836.25
14.144 2799236.25
14.148 2784631.25
14.153 2836975
14.157 2929848.5
14.161 3047099.75
14.165 3229735.75
14.169 3514579
14.174 3735556.75
14.178 3884972.25
14.182 4005098.25
14.186 4059827.5
14.19 3936117.75
14.195 3751759.5
14.199 3572535.25
14.203 3481697.5
14.207 3303875.75
14.211 3148019.75
14.216 3038785.25
14.22 2895151.5
14.224 2760836.75
14.228 2757395.75
14.232 2709340.25
14.237 2653216.25
14.241 2600169.5
14.245 2615854.5
14.249 2627863.25
14.253 2611416.5
14.257 2650642.25
14.262 2631094
14.266 2712698.5
14.27 2748653.5
14.274 2713270.5
14.278 2749962.75
14.283 2731537.5
14.287 2714302.75
14.291 2644595.25
14.295 2670998.75
14.299 2698652.5
14.304 2675573.25
14.308 2672091.5
14.312 2657900.75
14.316 2657075.5
14.32 2658682.25
14.325 2758849.5
14.329 2778141.5
14.333 2782186.75
14.337 2808101.5
14.341 2795050.5
14.346 2758384
14.35 2747699
14.354 2720603
14.358 2715343.75
14.362 2648766.5
14.366 2624246
14.371 2619086.25
14.375 2607791.5
14.379 2678771.75
14.383 2717953.75
14.387 2710216
14.392 2679261.25
14.396 2699853
14.4 2685911.25
14.404 2723512.75
14.408 2716588.75
14.413 2657250.25
14.417 2653217
14.421 2654861.5
14.425 2635971.75
14.429 2617930.25
14.434 2567317
14.438 2591851.5
14.442 2611194.5
14.446 2580797.5
14.45 2578422.75
14.455 2631393
14.459 2614833.75
14.463 2625246.75
14.467 2552827.25
14.471 2598573.25
14.475 2579115.75
14.48 2598482.25
14.484 2579239.75
14.488 2600206
14.492 2589633
14.496 2618681.25
14.501 2651785.5
14.505 2620721.5
14.509 2617593.75
14.513 2616853
14.517 2603442.25
14.522 2704455.75
14.526 2694329.5
14.53 2710211.75
14.534 2725900.25
14.538 2791307
14.543 2811764.5
14.547 2777574
14.551 2779421.5
14.555 2750717.75
14.559 2748890
14.564 2736734.25
14.568 2756493.5
14.572 2769387.75
14.576 2945832
14.58 3024348.25
14.584 3192485.75
14.589 3585018
14.593 3929168
14.597 4402568.5
14.601 4989001
14.605 5418055
14.61 5894054
14.614 6368543
14.618 6797750
14.622 6914822.5
14.626 7157674
14.631 7295133
14.635 7534686
14.639 7888773.5
14.643 8276874.5
14.647 8878965
14.652 9721505
14.656 10908157
14.66 12218110
14.664 13808803
14.668 15879459
14.673 18465600
14.677 21507194
14.681 24714544
14.685 27779328
14.689 30755798
14.693 33994328
14.698 37780464
14.702 41949908
14.706 45424188
14.71 47724916
14.714 48061256
14.719 45461876
14.723 40282432
14.727 34342068
14.731 29423456
14.735 25843188
14.74 22949056
14.744 20350706
14.748 17779166
14.752 15124291
14.756 12798824
14.761 10881734
14.765 9523264
14.769 8476720
14.773 7848989
14.777 7365264.5
14.782 6938281
14.786 6665732
14.79 6417649
14.794 6159873
14.798 5955058
14.802 5783645.5
14.807 5582915
14.811 5461653.5
14.815 5382835.5
14.819 5252148
14.823 5086139
14.828 4975788.5
14.832 4886187.5
14.836 4824916.5
14.84 4778192.5
14.844 4674040
14.849 4601748
14.853 4531098
14.857 4477743
14.861 4405997
14.865 4315598
14.87 4293476
14.874 4259967.5
14.878 4242092.5
14.882 4262183.5
14.886 4209535
14.891 4178190.5
14.895 4171041.5
14.899 4131916.5
14.903 4219824.5
14.907 4404954.5
14.912 4621717
14.916 5057549
14.92 5596492
14.924 6283939
14.928 6915307
14.932 7595509
14.937 7917821
14.941 7936322
14.945 7602872
14.949 7082071.5
14.953 6508467
14.958 5989100.5
14.962 5676176.5
14.966 5322786
14.97 5069636
14.974 4884354.5
14.979 4607558.5
14.983 4472689
14.987 4366391
14.991 4331744.5
14.995 4196182.5
15 4110473
15.004 4018156.5
15.008 3998365
15.012 3912429.25
15.016 3879496.5
15.021 3823439.75
15.025 3797529.75
15.029 3796909
15.033 3841184.75
15.037 3886776.5
15.041 3885227
15.046 3881626.25
15.05 3897725
15.054 3916307.25
15.058 3923543
15.062 3886607.25
15.067 3831852.75
15.071 3878266.75
15.075 3863438
15.079 3760573.5
15.083 3694206.75
15.088 3616551.25
15.092 3574437
15.096 3567239.25
15.1 3534763.5
15.104 3474360.25
15.109 3445395.75
15.113 3403364.5
15.117 3382839
15.121 3411449.5
15.125 3464885.25
15.13 3436032.25
15.134 3406937.25
15.138 3392344.25
15.142 3437599.5
15.146 3492019.75
15.15 3448871
15.155 3477771.75
15.159 3424135
15.163 3410487.75
15.167 3331775.5
15.171 3364108.25
15.176 3400553.5
15.18 3393785.25
15.184 3453354.5
15.188 3513083
15.192 3480802.75
15.197 3501812.25
15.201 3538019.25
15.205 3510605.25
15.209 3534643.5
15.213 3519752
15.218 3488217.5
15.222 3467893.75
15.226 3436023.5
15.23 3458348.25
15.234 3384904.25
15.239 3351989.75
15.243 3270586
15.247 3308960.25
15.251 3298929.25
15.255 3315415.25
15.259 3377332.75
15.264 3312125.25
15.268 3245741.25
15.272 3219705
15.276 3196456.5
15.28 3236613
15.285 3222937.25
15.289 3254375
15.293 3235917.25
15.297 3241554
15.301 3231371.25
15.306 3239911.75
15.31 3258670.75
15.314 3311215
15.318 3276148.25
15.322 3263782.25
15.327 3202029.25
15.331 3138676.25
15.335 3097524
15.339 3108535.5
15.343 3157370.5
15.348 3227892
15.352 3417055.5
15.356 3631150.25
15.36 3920284.25
15.364 4202409.5
15.368 4521183.5
15.373 4682407.5
15.377 4641329.5
15.381 4472824.5
15.385 4215024
15.389 3950961
15.394 3696763.5
15.398 3484459.25
15.402 3335969.5
15.406 3258600.25
15.41 3122484
15.415 3128397.75
15.419 3151297.75
15.423 3136858.75
15.427 3087944.25
15.431 3031418.25
15.436 3037923.75
15.44 2970105.5
15.444 3006776.75
15.448 3021585.5
15.452 3052403.75
15.457 2999411.75
15.461 3041122
15.465 3075886.75
15.469 3191451.5
15.473 3413647
15.477 3679443
15.482 4167764.25
15.486 4808325.5
15.49 5490402
15.494 6074074
15.498 6453191.5
15.503 6600554.5
15.507 6426457
15.511 5911271.5
15.515 5331042.5
15.519 4765629
15.524 4195312.5
15.528 3724121.75
15.532 3435117.5
15.536 3174152.75
15.54 3066841
15.545 3006909.5
15.549 2984090
15.553 3042236.25
15.557 3079539.75
15.561 3016355.75
15.566 3064101.25
15.57 3046375.75
15.574 3034378.5
15.578 3021183.75
15.582 2960355.25
15.586 2941818
15.591 2988386.25
15.595 2996701.25
15.599 2955550.75
15.603 2899300
15.607 2945515
15.612 2959502.75
15.616 2916349.5
15.62 2949858.25
15.624 3023597.75
15.628 3004142.5
15.633 3000660
15.637 3006958.5
15.641 2965880.75
15.645 2920027
15.649 2956954
15.654 2941965.25
15.658 2913486.75
15.662 2889261.5
15.666 2921383.25
15.67 2904795
15.675 2880791.75
15.679 2884287.25
15.683 2850946.75
15.687 2848777
15.691 2804338
15.696 2745465
15.7 2842387.5
15.704 2871306.5
15.708 2888469.5
15.712 2857239.75
15.716 2909362.5
15.721 2805856
15.725 2808183
15.729 2809721.25
15.733 2804139.75
15.737 2811724.5
15.742 2771078
15.746 2769162.5
15.75 2826775.25
15.754 2829439.25
15.758 2832474.5
15.763 2866830.75
15.767 2944558.75
15.771 2983450.5
15.775 2927194.25
15.779 3003406.75
15.784 3058693.5
15.788 3092317
15.792 3039677.25
15.796 3025116.75
15.8 3084055.75
15.805 3149885
15.809 3213656.75
15.813 3228383
15.817 3202320.5
15.821 3194511.5
15.825 3185865.5
15.83 3170930.25
15.834 3160307.5
15.838 3180744.75
15.842 3226505.5
15.846 3300033.5
15.851 3461092.25
15.855 3597609.75
15.859 3742683
15.863 3891039.5
15.867 3964303.25
15.872 3912832.5
15.876 3837634.75
15.88 3692404
15.884 3544174.75
15.888 3386480.75
15.893 3336421.5
15.897 3183676
15.901 3136997.5
15.905 3109093.25
15.909 3029506
15.914 3128032.5
15.918 3162051.5
15.922 3154569
15.926 3185259.5
15.93 3217071.75
15.934 3117659
15.939 3073023.5
15.943 3039481.5
15.947 2986464
15.951 2940048.25
15.955 2962445
15.96 3013048.25
15.964 3060034
15.968 3084632.75
15.972 3164383.75
15.976 3281837.75
15.981 3275739
15.985 3344112.25
15.989 3393044.5
15.993 3416620
15.997 3461806.5
16.002 3389586.75
16.006 3291674.5
16.01 3311012.5
16.014 3245938.25
16.018 3171915.25
16.023 3081968.5
16.027 3122685
16.031 3178193
16.035 3188750.5
16.039 3127805.25
16.043 3092777.75
16.048 3102118.75
16.052 3119038.75
16.056 3157400.25
16.06 3173063.25
16.064 3192225.5
16.069 3258074.75
16.073 3246095
16.077 3227402
16.081 3184156.25
16.085 3206433.25
16.09 3143884.25
16.094 3111873
16.098 3150126.25
16.102 3125184
16.106 3150749.25
16.111 3141784
16.115 3143889.5
16.119 3101727.25
16.123 3063712.75
16.127 3111448.75
16.132 3043791.75
16.136 3038447.25
16.14 3061939.5
16.144 3074925.75
16.148 2992101.75
16.152 3012467.75
16.157 3114601.75
16.161 3057369.25
16.165 3085860.25
16.169 3084120
16.173 3128263.5
16.178 3154480.5
16.182 3150801
16.186 3031571.5
16.19 3007470
16.194 3013498
16.199 3042617.75
16.203 2935318
16.207 2879858
16.211 2876345.25
16.215 2901156
16.22 2901850
16.224 2847575.5
16.228 2792512.75
16.232 2860979.25
16.236 2852864.25
16.241 2905219
16.245 2881667.25
16.249 2918035.25
16.253 2984619.75
16.257 3128285.5
16.261 3260289.25
16.266 3430369
16.27 3527957.5
16.274 3566130.75
16.278 3548961.5
16.282 3477859.75
16.287 3455414.25
16.291 3379759.25
16.295 3300289
16.299 3257017
16.303 3179364.75
16.308 3116550
16.312 3107430.25
16.316 3097353.5
16.32 3033652.75
16.324 3047742.25
16.329 3092017.25
16.333 3175874
16.337 3206981.5
16.341 3313513.25
16.345 3389049.5
16.35 3367113.75
16.354 3357019.75
16.358 3312455
16.362 3253698.5
16.366 3194725.25
16.37 3244438.5
16.375 3121697.5
16.379 3053599.25
16.383 2950282.25
16.387 2941947
16.391 2923266.25
16.396 2891478.25
16.4 2827762.25
16.404 2823691
16.408 2821514
16.412 2844739.25
16.417 2926703.25
16.421 2943960.75
16.425 2954040.5
16.429 2938022.75
16.433 2988439.75
16.438 2965470.5
16.442 2963324.75
16.446 2947570
16.45 2906182.25
16.454 2867351.75
16.459 2868093
16.463 2845258
16.467 2802510.75
16.471 2894618.25
16.475 2853330
16.48 2817076.75
16.484 2812809.75
16.488 2782294.75
16.492 2797833.5
16.496 2769761.25
16.5 2775062.5
16.505 2820978.75
16.509 2911413.75
16.513 2892781.5
16.517 2999878.5
16.521 3107838.75
16.526 3227708.5
16.53 3410160.75
16.534 3627987.25
16.538 3736007.25
16.542 3821159.25
16.547 3839460.25
16.551 3738302.75
16.555 3565061.25
16.559 3422302.5
16.563 3231342.25
16.568 3127500
16.572 3072322.5
16.576 3026086.25
16.58 3011442.75
16.584 3064856.25
16.589 3155143.75
16.593 3159112.5
16.597 3174072.5
16.601 3185519.75
16.605 3197481
16.609 3061107.25
16.614 3028739.75
16.618 3030243.75
16.622 2964138.75
16.626 2883930.5
16.63 2874005
16.635 2898059.75
16.639 2954004.75
16.643 2931304
16.647 2943397
16.651 2977534
16.656 2954491
16.66 2944202
16.664 2924035.75
16.668 2912309.5
16.672 2925030.75
16.677 2937049
16.681 2846847
16.685 2844209.75
16.689 2869509.75
16.693 2919254.75
16.698 2944855
16.702 3026367.75
16.706 3088990.25
16.71 3122901
16.714 3150960.5
16.718 3227458.25
16.723 3200168.5
16.727 3154748.75
16.731 3069016.75
16.735 3064636
16.739 3104855.75
16.744 3151090
16.748 3179159.5
16.752 3204564
16.756 3192606.75
16.76 3167496.75
16.765 3139952
16.769 3133765.25
16.773 3085137.25
16.777 3083645.25
16.781 2963235.5
16.786 2946886.5
16.79 2915740.5
16.794 2908539.25
16.798 2925534.75
16.802 2907501.5
16.807 3004506.25
16.811 2963579
16.815 2960894
16.819 2937976.25
16.823 2887838
16.827 2929285.25
16.832 2965926
16.836 2963913.25
16.84 2949037.25
16.844 2884741.75
16.848 2847333.5
16.853 2904335.25
16.857 2889266.5
16.861 2857568.25
16.865 2831033.75
16.869 2829222.5
16.874 2849736.5
16.878 2866271.75
16.882 2893364.5
16.886 2867812.25
16.89 2885986.5
16.895 2841364.5
16.899 2869338.75
16.903 2830393.75
16.907 2753359
16.911 2705390.5
16.916 2763556
16.92 2790674.25
16.924 2754113
16.928 2809852.5
16.932 2852769.25
16.936 2875158.25
16.941 2898511.5
16.945 2870856
16.949 2932775.75
16.953 2897929.25
16.957 2916012.5
16.962 2926019
16.966 2978030.25
16.97 3025203.5
16.974 2941063
16.978 2924054.5
16.983 2948071.25
16.987 2929515.75
16.991 2927291
16.995 2931272.75
16.999 2882712.5
17.004 2944314.75
17.008 2999733
17.012 2969673.75
17.016 3054785.25
17.02 3058308.5
17.025 3051810.5
17.029 3087936
17.033 3078406.75
17.037 3018455.25
17.041 2996211.5
17.045 3009924.25
17.05 3000035.25
17.054 3073780.25
17.058 3041924.5
17.062 3071763
17.066 3008662.75
17.071 3015013.5
17.075 2948683
17.079 3003891.75
17.083 3024545
17.087 2986555.75
17.092 2945410.75
17.096 2925106.75
17.1 2899857.5
17.104 2896126.25
17.108 2916538.5
17.113 2910519.25
17.117 2912757.25
17.121 2868081
17.125 2888550.25
17.129 2896650.5
17.134 2921843.5
17.138 2936037.25
17.142 2881954
17.146 2905557.75
17.15 2961512.75
17.154 2995827.5
17.159 2980679
17.163 2966597
17.167 2963308
17.171 2961266.25
17.175 2970812.75
17.18 2973220
17.184 2952433
17.188 3087587.25
17.192 3106032.75
17.196 3088440.25
17.201 3170077.75
17.205 3301878.25
17.209 3416002
17.213 3453832.5
17.217 3466807
17.222 3446906.75
17.226 3418645.25
17.23 3422783.75
17.234 3482509
17.238 3541575.75
17.243 3589172.75
17.247 3546983.75
17.251 3505278
17.255 3502160.75
17.259 3506419
17.263 3518748.5
17.268 3548904.25
17.272 3596110.75
17.276 3592263.75
17.28 3568422.5
17.284 3540999.5
17.289 3486478
17.293 3527058.25
17.297 3597043.75
17.301 3599013.25
17.305 3739703.5
17.31 3943537.5
17.314 4093215
17.318 4375115
17.322 4483294
17.326 4389603.5
17.331 4323786
17.335 4121763
17.339 3980954.75
17.343 3796998
17.347 3643803
17.352 3555037.75
17.356 3481044.5
17.36 3493469.25
17.364 3444686.5
17.368 3380302.25
17.373 3363209.25
17.377 3358316.5
17.381 3309539.75
17.385 3188251.25
17.389 3227848.5
17.393 3205328.5
17.398 3079643
17.402 3007282.25
17.406 2956586.25
17.41 2959217.75
17.414 2989918.5
17.419 3007353
17.423 3045588.25
17.427 3015402
17.431 3044814.25
17.435 3087308.75
17.44 3074296
17.444 2996823
17.448 3014698.5
17.452 3027406.5
17.456 2986577.5
17.461 2978266.25
17.465 2936218
17.469 2977372.25
17.473 2976712.75
17.477 2944091
17.482 2964484.5
17.486 2937858.75
17.49 2920105.75
17.494 2891819.75
17.498 2896849
17.502 2960434
17.507 2961974.5
17.511 3004510
17.515 2987531.25
17.519 2984881.5
17.523 3014097
17.528 3023471.75
17.532 3011713.25
17.536 3027609.25
17.54 3036111.5
17.544 3048834.75
17.549 2992709.75
17.553 2943264.75
17.557 2936745.25
17.561 2979455.5
17.565 3093719.5
17.57 3175963.25
17.574 3278217.5
17.578 3309583.75
17.582 3298180.75
17.586 3346713
17.591 3405373
17.595 3387646.75
17.599 3320403.25
17.603 3316680.25
17.607 3248881.25
17.611 3237364.5
17.616 3128233.25
17.62 3145002
17.624 3102960
17.628 3085199.25
17.632 3081509.25
17.637 3083541.75
17.641 3058789.25
17.645 3102282.5
17.649 3129844.25
17.653 3151801.5
17.658 3159625.5
17.662 3310291.75
17.666 3434613.25
17.67 3697682
17.674 3997177.25
17.679 4164555.25
17.683 4196976.5
17.687 4174405.25
17.691 4105729.75
17.695 3938785.5
17.7 3681633.25
17.704 3546497.75
17.708 3366367.75
17.712 3254848.25
17.716 3211776.5
17.72 3120133
17.725 3195909
17.729 3119451
17.733 3095303.25
17.737 3043203.75
17.741 3064327.75
17.746 3055518.5
17.75 3026563.75
17.754 3077281
17.758 3024746
17.762 3056789.75
17.767 3092556
17.771 3113583.75
17.775 3083806
17.779 3078045.25
17.783 3194373.25
17.788 3321066.25
17.792 3474178.75
17.796 3681080
17.8 3949630.25
17.804 4249393
17.809 4476332
17.813 4591660
17.817 4754061.5
17.821 4690255.5
17.825 4577210.5
17.829 4317673.5
17.834 4036197
17.838 3706260.5
17.842 3468155.25
17.846 3284589.5
17.85 3166358
17.855 3053650.5
17.859 3110557.75
17.863 3035796.25
17.867 3037208.5
17.871 3081842.5
17.876 3053765
17.88 3118591.25
17.884 3210639.5
17.888 3152801.5
17.892 3235997.5
17.897 3262243.75
17.901 3265075.25
17.905 3313338.25
17.909 3346788.25
17.913 3326842.25
17.918 3308248.25
17.922 3267000
17.926 3250403.5
17.93 3223811.75
17.934 3209598
17.938 3174100.75
17.943 3162186
17.947 3099098.25
17.951 3101707.5
17.955 3096585.75
17.959 3088423
17.964 3084907.25
17.968 3098556
17.972 3154650.75
17.976 3157532
17.98 3293452.25
17.985 3330329
17.989 3440732.5
17.993 3433918
17.997 3453904.25
18.001 3475707.75
18.006 3426426.5
18.01 3358896
18.014 3406278.5
18.018 3499292
18.022 3628963
18.027 3642502.75
18.031 3691581
18.035 3673127
18.039 3650685.5
18.043 3536631.75
18.048 3426506.25
18.052 3354724.25
18.056 3311483.25
18.06 3286661.5
18.064 3228074
18.068 3231915
18.073 3201895.5
18.077 3224235.75
18.081 3224226.75
18.085 3185095.25
18.089 3193767.25
18.094 3152554.5
18.098 3116425.25
18.102 3054798
18.106 3139527.25
18.11 3083933.75
18.115 3055374.5
18.119 3004299.5
18.123 3026327.25
18.127 3077435
18.131 3044108
18.136 3053366.25
18.14 3052494.5
18.144 3038123.75
18.148 3042454.5
18.152 3044544.5
18.157 3044278.25
18.161 2989689
18.165 2979761.25
18.169 2967075.25
18.173 3036544.5
18.177 2948135
18.182 3041852.5
18.186 3064427.75
18.19 3116117.25
18.194 3194880.75
18.198 3222162.25
18.203 3241133.5
18.207 3317235
18.211 3275386.25
18.215 3224070.75
18.219 3125686.5
18.224 3148296.25
18.228 3112036.25
18.232 3075579.5
18.236 3019408.25
18.24 3012571
18.245 3016744
18.249 2968614
18.253 2975938.75
18.257 2956927
18.261 3001223
18.266 3019311.5
18.27 2991023.5
18.274 3015812.5
18.278 3020460.25
18.282 3033767.25
18.286 2971425.25
18.291 2989742
18.295 3014782
18.299 3004572.5
18.303 2978823.75
18.307 2976571.75
18.312 2894968.75
18.316 2958136.75
18.32 2974411
18.324 3003396
18.328 2994532.75
18.333 3002931.25
18.337 2989963
18.341 3013866
18.345 3015330.25
18.349 3070995.5
18.354 3087896
18.358 3036485
18.362 3024453.25
18.366 3054606
18.37 3045707.75
18.375 3016997.25
18.379 3060649
18.383 3085846.25
18.387 3111934
18.391 3111439
18.395 3093520.5
18.4 3173381.5
18.404 3170451.75
18.408 3200443.5
18.412 3191109
18.416 3234062.5
18.421 3277715.5
18.425 3262172
18.429 3281429.5
18.433 3201309.75
18.437 3173675.75
18.442 3157700.25
18.446 3154295.5
18.45 3149967.25
18.454 3159570.75
18.458 3187703
18.463 3124442.5
18.467 3072898.75
18.471 3141691.75
18.475 3180152.25
18.479 3214775.5
18.484 3166281.75
18.488 3186587.5
18.492 3168913.5
18.496 3157973.75
18.5 3150254.25
18.504 3172327.25
18.509 3107547
18.513 3164187.75
18.517 3112606.25
18.521 3151469
18.525 3133360.5
18.53 3088888.25
18.534 3075003.75
18.538 3185881.5
18.542 3237364.25
18.546 3215059.5
18.551 3225287
18.555 3209376.5
18.559 3247372
18.563 3311158.75
18.567 3325590.5
18.572 3292391.5
18.576 3370901.75
18.58 3541233.5
18.584 3721369.25
18.588 3810366
18.593 3933141.75
18.597 4045421
18.601 4070016.5
18.605 3975874.5
18.609 3895651.25
18.613 3706608.25
18.618 3545983.75
18.622 3433660.75
18.626 3362361.5
18.63 3270712.5
18.634 3230854.75
18.639 3191535.5
18.643 3205858.5
18.647 3221486.75
18.651 3194382.25
18.655 3177697.75
18.66 3242597
18.664 3291354.5
18.668 3262911
18.672 3263537.5
18.676 3247719.75
18.681 3224647.75
18.685 3354187.5
18.689 3339489.75
18.693 3394764
18.697 3445272.25
18.702 3567128.75
18.706 3560454.5
18.71 3523588.25
18.714 3480439.5
18.718 3498796.25
18.723 3534128
18.727 3532954
18.731 3505848
18.735 3450284.75
18.739 3410740.75
18.743 3453971
18.748 3480468
18.752 3481460.25
18.756 3547300
18.76 3554878.5
18.764 3713766.25
18.769 3891158.75
18.773 4073245
18.777 4200027.5
18.781 4383368.5
18.785 4457127
18.79 4516451
18.794 4499974.5
18.798 4405427
18.802 4278207
18.806 4134495.5
18.811 3918251.5
18.815 3736518
18.819 3648901
18.823 3573193
18.827 3559406.25
18.832 3517588
18.836 3462824.75
18.84 3468428.5
18.844 3496830.75
18.848 3433363.25
18.852 3396582.25
18.857 3447742.75
18.861 3493221
18.865 3440775
18.869 3517753.25
18.873 3611611.75
18.878 3641694.25
18.882 3594402.75
18.886 3602154.75
18.89 3585391.25
18.894 3594624
18.899 3485696.75
18.903 3412087.5
18.907 3335676.5
18.911 3353029.75
18.915 3276852
18.92 3270744.5
18.924 3254519
18.928 3269899
18.932 3230987.25
18.936 3243452
18.941 3283752.25
18.945 3235628.5
18.949 3283860.25
18.953 3311556.5
18.957 3231253.5
18.961 3186423.5
18.966 3222904.75
18.97 3237904.5
18.974 3226048.75
18.978 3183183.25
18.982 3167439
18.987 3235988.75
18.991 3183347.75
18.995 3187030
18.999 3145531
19.003 3183744.75
19.008 3219580.25
19.012 3222173
19.016 3203716.5
19.02 3154147.25
19.024 3242148
19.029 3165387.5
19.033 3144441.25
19.037 3158523.25
19.041 3202825.75
19.045 3140954.75
19.05 3116217.75
19.054 3048699.5
19.058 3074507
19.062 3124761.25
19.066 3158703
19.07 3155768
19.075 3127723.25
19.079 3199065.5
19.083 3210591.75
19.087 3169623.75
19.091 3235533.75
19.096 3277825.5
19.1 3400809.75
19.104 3484714.5
19.108 3571838.5
19.112 3645919.5
19.117 3693875.25
19.121 3644563
19.125 3629320.25
19.129 3559245.5
19.133 3526487.75
19.138 3501289
19.142 3450483.5
19.146 3455804.5
19.15 3535386.25
19.154 3540018.75
19.159 3511670.5
19.163 3502787.5
19.167 3534257.75
19.171 3435196.75
19.175 3384584.75
19.179 3415619.25
19.184 3353226.25
19.188 3356234.75
19.192 3260145.75
19.196 3245259
19.2 3199181.25
19.205 3195426
19.209 3219703
19.213 3242766.75
19.217 3280479.25
19.221 3193452.75
19.226 3175349.75
19.23 3214391.75
19.234 3262131
19.238 3260231.5
19.242 3302588.25
19.247 3351439.5
19.251 3325625.5
19.255 3293618.25
19.259 3294231.25
19.263 3315074
19.268 3334434
19.272 3328880
19.276 3285796.75
19.28 3291115
19.284 3289019
19.288 3331628.75
19.293 3335944.5
19.297 3330860.75
19.301 3312113.75
19.305 3280374.25
19.309 3297837.25
19.314 3234766.25
19.318 3257122.75
19.322 3220195
19.326 3278206.25
19.33 3303023.5
19.335 3342844.5
19.339 3452212.5
19.343 3362978.25
19.347 3415945
19.351 3416968.5
19.356 3472561.75
19.36 3526521.75
19.364 3597856.5
19.368 3742287.25
19.372 3898334.75
19.377 4098939.5
19.381 4252610.5
19.385 4365635
19.389 4443715.5
19.393 4554335.5
19.398 4540953.5
19.402 4369077.5
19.406 4158112.5
19.41 3935601.75
19.414 3795796.25
19.418 3621715.75
19.423 3543387
19.427 3457528.25
19.431 3500331.75
19.435 3459887
19.439 3422109.25
19.444 3382480
19.448 3321958.75
19.452 3343121
19.456 3400707.75
19.46 3424632.5
19.465 3326169.5
19.469 3303933.25
19.473 3284626
19.477 3320547
19.481 3334056.25
19.486 3334258.75
19.49 3382836.75
19.494 3420301.75
19.498 3510881.5
19.502 3565924.25
19.507 3545710.25
19.511 3623039.25
19.515 3640505.75
19.519 3619158
19.523 3691724.75
19.527 3699861.25
19.532 3752850.75
19.536 3828745
19.54 3888521
19.544 3970894.75
19.548 4086267.25
19.553 4050944
19.557 4117318.75
19.561 4157678.5
19.565 4137799.25
19.569 4062961
19.574 4035746.75
19.578 4028410.25
19.582 3975664.75
19.586 3847154.5
19.59 3747901.75
19.595 3674038.5
19.599 3637191.5
19.603 3618423.75
19.607 3695948.5
19.611 3698828.75
19.616 3671074.25
19.62 3621779.25
19.624 3599355.25
19.628 3595190.5
19.632 3492948.75
19.636 3527804
19.641 3489612.75
19.645 3448982
19.649 3460268
19.653 3398867.75
19.657 3415985.25
19.662 3475099
19.666 3420866
19.67 3359966.25
19.674 3412334.5
19.678 3458337.25
19.683 3430161.25
19.687 3440864.5
19.691 3428537.5
19.695 3415850.5
19.699 3431638.5
19.704 3467276.5
19.708 3377475.75
19.712 3341360.25
19.716 3349070.25
19.72 3409947.5
19.725 3422202.25
19.729 3418422.25
19.733 3410551.5
19.737 3410837
19.741 3359542.5
19.745 3297384.25
19.75 3328672.25
19.754 3359935.5
19.758 3390331.75
19.762 3388558
19.766 3484644.5
19.771 3423672.25
19.775 3448878.25
19.779 3442910
19.783 3447178.75
19.787 3472364
19.792 3508893.25
19.796 3627642.75
19.8 3638211.25
19.804 3699588.25
19.808 3743886.5
19.813 3869084.5
19.817 3945496.25
19.821 4131652.5
19.825 4221503
19.829 4194402.5
19.834 4217568.5
19.838 4120245.5
19.842 4116793.25
19.846 4079905.75
19.85 4169448.25
19.854 4231520
19.859 4346654
19.863 4278539
19.867 4308915.5
19.871 4269006.5
19.875 4116572.75
19.88 4005070.25
19.884 3836718.75
19.888 3830244.25
19.892 3722774.5
19.896 3683386.75
19.901 3607277
19.905 3521229.25
19.909 3477235.75
19.913 3502116.5
19.917 3563421.75
19.922 3522388.25
19.926 3571348.5
19.93 3563571.75
19.934 3541965.75
19.938 3587934
19.943 3588952.75
19.947 3511003.75
19.951 3516016.25
19.955 3479703
19.959 3485328.25
19.964 3442182
19.968 3519011.5
19.972 3505069.25
19.976 3495146.5
19.98 3504305.5
19.984 3489561.5
19.989 3494976.5
19.993 3459546
19.997 3488200.5
20.001 3624463
20.005 3745601
20.01 3780805
20.014 3836826.25
20.018 3913237.75
20.022 3963859.5
20.026 4005061.5
20.031 4031075.75
20.035 4050587.25
20.039 4017211.5
20.043 3897930.75
20.047 3910777.75
20.052 3831230.75
20.056 3812659.5
20.06 3833631.75
20.064 3839418
20.068 3798920
20.073 3866175.75
20.077 3897393.25
20.081 3928775
20.085 3956056.75
20.089 4050561.5
20.093 4034027.25
20.098 3937416
20.102 3767715.75
20.106 3694360.75
20.11 3723250.25
20.114 3765450.75
20.119 3724103.25
20.123 3669157.25
20.127 3673559.5
20.131 3674579.75
20.135 3626031
20.14 3637169
20.144 3635308.5
20.148 3638877.75
20.152 3583700
20.156 3619163.75
20.161 3520038.75
20.165 3541268.25
20.169 3536784.25
20.173 3525500.25
20.177 3521637.25
20.182 3535484.75
20.186 3579153.5
20.19 3609126.25
20.194 3610574.25
20.198 3655615.75
20.202 3610682.5
20.207 3629607.75
20.211 3637768.75
20.215 3699794.25
20.219 3721415
20.223 3627898.5
20.228 3609618.75
20.232 3625391
20.236 3719010
20.24 3627338.75
20.244 3604796.5
20.249 3609493.5
20.253 3543866.5
20.257 3603293.5
20.261 3598782.5
20.265 3581273.75
20.27 3605192.5
20.274 3629535.5
20.278 3582220.5
20.282 3589704
20.286 3500514.75
20.291 3576596.5
20.295 3545878.5
20.299 3517740.75
20.303 3482851.5
20.307 3500199.75
20.311 3503421
20.316 3555631.25
20.32 3552379.75
20.324 3520417
20.328 3487995.5
20.332 3570278.75
20.337 3563301.75
20.341 3461584
20.345 3490479
20.349 3513027
20.353 3534431
20.358 3520815.25
20.362 3503606.25
20.366 3513907.75
20.37 3576043.75
20.374 3690641
20.379 3676356.25
20.383 3623093
20.387 3571950
20.391 3565069
20.395 3554644.5
20.4 3541370.75
20.404 3570599.25
20.408 3580688.25
20.412 3569538.75
20.416 3592736.75
20.42 3632052
20.425 3599444
20.429 3639523
20.433 3606476
20.437 3593705.75
20.441 3655766
20.446 3665252
20.45 3773376.25
20.454 3737971.25
20.458 3745735
20.462 3794869
20.467 3772248.5
20.471 3745513
20.475 3800576
20.479 3845915
20.483 3902829.25
20.488 3843620
20.492 3778380.25
20.496 3799512.25
20.5 3790517.25
20.504 3937900.75
20.509 3854038.75
20.513 3746674.75
20.517 3809182.5
20.521 3781341
20.525 3780335.25
20.53 3794709.75
20.534 3783152.25
20.538 3741427.25
20.542 3779084
20.546 3769635.75
20.55 3762696
20.555 3855468.5
20.559 3838028
20.563 3877793.75
20.567 3837485.5
20.571 3890557
20.576 3876748
20.58 3880247.75
20.584 3817345
20.588 3811228.25
20.592 3923140
20.597 3908858
20.601 3907861.75
20.605 3918486.25
20.609 3947924.25
20.613 3894202.25
20.618 3957252.25
20.622 3912613
20.626 3882492.75
20.63 3864984.5
20.634 3857488
20.639 3873555.75
20.643 3953984
20.647 3964455
20.651 4011019
20.655 4053760.75
20.659 4084631.25
20.664 4093383.25
20.668 4106541
20.672 4108460
20.676 4097602.25
20.68 4112634.25
20.685 3970399.75
20.689 3914106.25
20.693 3859777.5
20.697 3826178.25
20.701 3848584
20.706 3740845
20.71 3785207.75
20.714 3771888.5
20.718 3878699.25
20.722 3868804.25
20.727 3901564.5
20.731 3970713.75
20.735 3962197
20.739 3997838.25
20.743 3994984.75
20.748 4049156.5
20.752 4074082
20.756 4144975
20.76 4152520.25
20.764 4221546
20.768 4309463
20.773 4513897.5
20.777 4503344
20.781 4489029.5
20.785 4360588
20.789 4296285
20.794 4234149
20.798 4188320.25
20.802 4094930.75
20.806 4027828.25
20.81 3972553
20.815 4013610
20.819 3978418
20.823 4007847
20.827 3880416.5
20.831 3912341
20.836 3965243.75
20.84 3949257.75
20.844 3890196.5
20.848 3928887.75
20.852 3978537
20.857 3965117.25
20.861 3960609.5
20.865 3971053
20.869 3982480.5
20.873 4065353.5
20.877 4078159
20.882 4132780
20.886 4318951
20.89 4372857.5
20.894 4448697
20.898 4523823.5
20.903 4598179
20.907 4687301
20.911 4680349.5
20.915 4600661
20.919 4532802
20.924 4425172.5
20.928 4335593.5
20.932 4256220
20.936 4189435
20.94 4180148.5
20.945 4106481.75
20.949 4126263.5
20.953 4103339.5
20.957 4041613.75
20.961 4016568
20.966 4016337.25
20.97 4010025
20.974 4038525.75
20.978 4123305.75
20.982 4070229.75
20.987 4030991.25
20.991 3991082.75
20.995 4006342.25
20.999 4016836.5
21.003 4012569.75
21.007 4097816.25
21.012 4166036.75
21.016 4217027.5
21.02 4336385.5
21.024 4386729.5
21.028 4500474.5
21.033 4491559
21.037 4472045
21.041 4475763.5
21.045 4433632.5
21.049 4338730
21.054 4306782
21.058 4289395
21.062 4216617
21.066 4142714
21.07 4080796.25
21.075 4044489.75
21.079 4027892.25
21.083 4044270.25
21.087 4062026
21.091 4007343.25
21.096 3958240
21.1 3961979.25
21.104 3979125.25
21.108 4047657.5
21.112 4047262.25
21.116 4070035.25
21.121 4023475.5
21.125 4041779.5
21.129 4074836.5
21.133 4057387.5
21.137 3988845
21.142 4014479
21.146 4039846
21.15 4019280
21.154 4022752.25
21.158 4047955.25
21.163 3950042.5
21.167 3889997.25
21.171 3888777.5
21.175 3966876.5
21.179 3957555.75
21.184 3939909.25
21.188 3997407
21.192 4006863.75
21.196 3991495.75
21.2 4003820.75
21.205 3945511.5
21.209 3932276.5
21.213 3958807
21.217 4019320
21.221 4074450.5
21.225 4115245
21.23 4123596
21.234 4133723.75
21.238 4130438.75
21.242 4134863.75
21.246 4164861.5
21.251 4141261.75
21.255 4144065
21.259 4124806.5
21.263 4204057
21.267 4200656.5
21.272 4223310
21.276 4176712
21.28 4150269.5
21.284 4136614.5
21.288 4202313.5
21.293 4162045.75
21.297 4160067.25
21.301 4243689
21.305 4230761
21.309 4236314
21.314 4283056.5
21.318 4221070.5
21.322 4238189
21.326 4264789.5
21.33 4244903.5
21.334 4275351.5
21.339 4360373
21.343 4430080
21.347 4328495
21.351 4369009
21.355 4363768
21.36 4467900.5
21.364 4469215.5
21.368 4410900.5
21.372 4402839.5
21.376 4410359
21.381 4477443
21.385 4383578.5
21.389 4415395
21.393 4399577.5
21.397 4521587.5
21.402 4453745
21.406 4511443.5
21.41 4586111.5
21.414 4653427
21.418 4623603.5
21.423 4660408
21.427 4787786.5
21.431 4835837
21.435 4929377.5
21.439 4960146.5
21.444 4973240
21.448 4990869
21.452 5033534.5
21.456 5061647
21.46 5104681.5
21.464 5114497
21.469 5081866
21.473 5087221
21.477 5192654.5
21.481 5320179
21.485 5404591.5
21.49 5480225
21.494 5541961.5
21.498 5576376.5
21.502 5597910.5
21.506 5584575.5
21.511 5651307.5
21.515 5636295
21.519 5685288
21.523 5760390
21.527 5767232
21.532 5834383
21.536 5835674.5
21.54 5847980.5
21.544 5892730.5
21.548 6036812
21.553 5955958
21.557 5979853
21.561 5935501.5
21.565 6112748.5
21.569 6054189
21.573 6137599.5
21.578 6237994.5
21.582 6196171.5
21.586 6279191.5
21.59 6292019.5
21.594 6381616
21.599 6348690
21.603 6512731
21.607 6454480
21.611 6408562.5
21.615 6314344.5
21.62 6251467
21.624 6352489
21.628 6358039
21.632 6377823
21.636 6223277
21.641 6191344
21.645 6248542.5
21.649 6250701.5
21.653 6103821.5
21.657 6072403
21.662 5956628.5
21.666 5870734
21.67 5700422
21.674 5698127
21.678 5679201
21.682 5664830
21.687 5534821.5
21.691 5449142
21.695 5426076
21.699 5399951.5
21.703 5373007
21.708 5369286.5
21.712 5406247.5
21.716 5429854
21.72 5400285
21.724 5297912.5
21.729 5277704.5
21.733 5523278.5
21.737 5772548.5
21.741 6184727
21.745 6780316.5
21.75 7575425
21.754 8557112
21.758 9615152
21.762 10745319
21.766 11445446
21.771 11534667
21.775 11081042
21.779 10388578
21.783 9501068
21.787 8540172
21.791 7771892
21.796 7196237
21.8 6781078
21.804 6426116.5
21.808 6258072.5
21.812 6112757.5
21.817 5948150.5
21.821 5850283.5
21.825 5801129.5
21.829 5782656.5
21.833 5716390.5
21.838 5644702.5
21.842 5619643
21.846 5530396
21.85 5462470.5
21.854 5512903.5
21.859 5472396
21.863 5359283.5
21.867 5272132.5
21.871 5225418.5
21.875 5276332
21.88 5251979
21.884 5217826
21.888 5286252
21.892 5371665.5
21.896 5464515
21.9 5488666.5
21.905 5597188
21.909 5591643.5
21.913 5565286.5
21.917 5424190
21.921 5394237.5
21.926 5249768
21.93 5186241
21.934 5070762
21.938 5045979
21.942 4963685
21.947 4948732
21.951 5038789
21.955 5037590
21.959 5051671
21.963 4956964.5
21.968 4984110
21.972 4866302
21.976 4787546
21.98 4829849
21.984 4917334
21.989 4896270
21.993 4857737.5
21.997 4858814
22.001 4855115.5
22.005 4791141.5
22.01 4750049
22.014 4755624.5
22.018 4831044.5
22.022 4812503.5
22.026 4782194
22.03 4757254
22.035 4820031
22.039 4807984
22.043 4822910
22.047 4776126.5
22.051 4804999.5
22.056 4786900.5
22.06 4846971.5
22.064 4876415.5
22.068 4999442
22.072 4994210.5
22.077 5011443
22.081 5030905.5
22.085 4935213.5
22.089 4978539.5
22.093 4988917
22.098 4884916.5
22.102 4818943.5
22.106 4775809.5
22.11 4766199.5
22.114 4740227
22.119 4757195
22.123 4775335
22.127 4745383
22.131 4715569
22.135 4716438.5
22.139 4719583.5
22.144 4789235
22.148 4855220
22.152 4863354
22.156 4981324
22.16 4930364
22.165 4940297
22.169 4979807
22.173 5001800.5
22.177 5093197.5
22.181 5114395.5
22.186 5217548.5
22.19 5166671
22.194 5081937
22.198 5006956.5
22.202 5036067
22.207 5073552.5
22.211 5036684.5
22.215 5105162.5
22.219 5241759
22.223 5405348
22.228 5851633.5
22.232 6523101.5
22.236 7392939
22.24 8432453
22.244 9689135
22.248 10971934
22.253 11842919
22.257 12436590
22.261 12402454
22.265 11939071
22.269 10879534
22.274 9599935
22.278 8409384
22.282 7319644
22.286 6563839.5
22.29 5967109.5
22.295 5532497
22.299 5345895.5
22.303 5258227.5
22.307 5232089.5
22.311 5197450.5
22.316 5173908.5
22.32 5051027
22.324 5073195.5
22.328 5094944
22.332 5153478.5
22.337 5136929
22.341 5155176.5
22.345 5201357
22.349 5138067.5
22.353 5112667.5
22.358 5132768
22.362 5127124.5
22.366 4956233
22.37 4957876.5
22.374 4950802.5
22.378 4886607.5
22.383 4863905.5
22.387 4882846.5
22.391 4850525.5
22.395 4923025
22.399 5044204
22.404 5090056.5
22.408 5099442
22.412 5172255.5
22.416 5162402
22.42 5277888
22.425 5320888
22.429 5247109
22.433 5310981.5
22.437 5272129.5
22.441 5220480
22.446 5152127
22.45 5134282.5
22.454 5077998
22.458 5123246.5
22.462 5011372.5
22.467 5051446.5
22.471 5039877.5
22.475 4988046
22.479 5046050.5
22.483 5034266.5
22.487 5121098
22.492 5068183.5
22.496 5022878
22.5 4961016.5
22.504 4941134
22.508 5048629.5
22.513 5072051
22.517 5003309.5
22.521 4970647.5
22.525 4989151.5
22.529 4964831
22.534 5027937
22.538 4963134.5
22.542 5026429
22.546 4995851.5
22.55 4998694.5
22.555 5002693.5
22.559 5095674
22.563 5120755.5
22.567 5084049.5
22.571 5121951
22.576 5182309.5
22.58 5222746
22.584 5202364.5
22.588 5251141.5
22.592 5128058.5
22.596 5128597
22.601 5168684
22.605 5184622
22.609 5122757
22.613 5073650.5
22.617 4971255.5
22.622 4975248.5
22.626 5052664.5
22.63 5087225
22.634 5162040
22.638 5188040.5
22.643 5053046.5
22.647 4938814.5
22.651 4984159
22.655 5028772
22.659 5048495
22.664 5089656
22.668 5066527.5
22.672 5075278.5
22.676 5079655.5
22.68 5063807
22.685 5073834
22.689 5037637.5
22.693 5120727.5
22.697 5088679.5
22.701 5154251
22.705 5173754
22.71 5118123.5
22.714 5114378
22.718 5054103.5
22.722 5066757.5
22.726 5104078
22.731 5097529.5
22.735 5092568
22.739 5059006
22.743 5098556
22.747 5101398.5
22.752 5116275
22.756 5093146.5
22.76 4987097
22.764 5043581.5
22.768 5051554.5
22.773 5123567
22.777 5115724.5
22.781 5097488
22.785 5039430
22.789 5020498.5
22.794 5054667.5
22.798 5191607
22.802 5316788
22.806 5330726.5
22.81 5492970
22.815 5576899.5
22.819 5569547.5
22.823 5606992
22.827 5678207
22.831 5704073
22.835 5730288
22.84 5687151
22.844 5611368.5
22.848 5557543
22.852 5606699
22.856 5619504
22.861 5662180
22.865 5656765
22.869 5731347
22.873 5824211.5
22.877 5975008.5
22.882 6060292.5
22.886 6086990.5
22.89 6203307.5
22.894 6328430.5
22.898 6515165.5
22.903 6605140.5
22.907 6520905.5
22.911 6392218.5
22.915 6296607.5
22.919 6123129.5
22.924 6000660
22.928 5909622
22.932 5915385
22.936 5756831
22.94 5667267
22.944 5604529.5
22.949 5566239
22.953 5539796
22.957 5486591.5
22.961 5404547.5
22.965 5379933
22.97 5356834
22.974 5324992
22.978 5362913
22.982 5336704
22.986 5314838.5
22.991 5369035.5
22.995 5456983
22.999 5455502
23.003 5531852.5
23.007 5502103.5
23.012 5565827.5
23.016 5629014.5
23.02 5700169.5
23.024 5809214
23.028 5798288
23.033 5694055.5
23.037 5670173.5
23.041 5636980.5
23.045 5652256.5
23.049 5592211.5
23.053 5437617.5
23.058 5437075.5
23.062 5535257.5
23.066 5520971
23.07 5443074.5
23.074 5425084.5
23.079 5485025
23.083 5500173.5
23.087 5425480.5
23.091 5370529.5
23.095 5380505
23.1 5399150.5
23.104 5296703
23.108 5275706
23.112 5390311.5
23.116 5394812.5
23.121 5356010.5
23.125 5365903.5
23.129 5319440.5
23.133 5266793
23.137 5284324.5
23.142 5264678
23.146 5288006
23.15 5311853
23.154 5299975.5
23.158 5358867.5
23.162 5374810
23.167 5255712.5
23.171 5354842.5
23.175 5369836.5
23.179 5395366.5
23.183 5338301.5
23.188 5360079.5
23.192 5390109.5
23.196 5339126.5
23.2 5446836.5
23.204 5465807
23.209 5394264
23.213 5463969
23.217 5467159
23.221 5540717.5
23.225 5498782.5
23.23 5493197
23.234 5392383
23.238 5419482
23.242 5500846
23.246 5490824.5
23.251 5508229
23.255 5437686
23.259 5457434
23.263 5465074
23.267 5467773
23.272 5451388
23.276 5436334
23.28 5474319
23.284 5536935.5
23.288 5576811
23.292 5640609
23.297 5723334
23.301 5767772
23.305 5744338
23.309 5717602
23.313 5757403.5
23.318 5728419
23.322 5692750
23.326 5703516.5
23.33 5623595
23.334 5573256.5
23.339 5531203
23.343 5508601
23.347 5518293.5
23.351 5497336
23.355 5514390
23.36 5577213.5
23.364 5568456.5
23.368 5528800
23.372 5568783
23.376 5642323.5
23.381 5719186
23.385 5765590
23.389 5721462
23.393 5852631.5
23.397 5807254.5
23.401 5774424
23.406 5664170.5
23.41 5701211
23.414 5790346
23.418 5739330.5
23.422 5650094.5
23.427 5574776.5
23.431 5550204.5
23.435 5558215
23.439 5514889.5
23.443 5535188
23.448 5631343.5
23.452 5613290.5
23.456 5540569.5
23.46 5616781
23.464 5693530
23.469 5691923
23.473 5715870.5
23.477 5722135.5
23.481 5788757
23.485 5894125.5
23.49 5968177
23.494 6045312
23.498 5975454
23.502 5909187.5
23.506 5890734
23.51 5829426.5
23.515 5860777.5
23.519 5859974
23.523 5885891
23.527 5870645
23.531 5840982.5
23.536 5741017.5
23.54 5639410.5
23.544 5638888
23.548 5680815
23.552 5742477.5
23.557 5698899
23.561 5715109.5
23.565 5749818
23.569 5788007.5
23.573 5761025.5
23.578 5740450.5
23.582 5848773
23.586 5877549
23.59 5813583.5
23.594 5849547
23.599 5884889.5
23.603 5922759.5
23.607 5746413.5
23.611 5746594
23.615 5805186
23.619 5760321.5
23.624 5768766.5
23.628 5742202
23.632 5760016
23.636 5822137.5
23.64 5859177.5
23.645 5831794.5
23.649 5904380
23.653 5811752
23.657 5776588.5
23.661 5902363
23.666 5909666.5
23.67 5849595
23.674 5827876
23.678 5869708.5
23.682 5902059.5
23.687 5871622.5
23.691 5883677
23.695 5814080
23.699 5805133.5
23.703 5779863.5
23.708 5830404.5
23.712 5922342
23.716 5871666.5
23.72 5869417
23.724 5967462
23.729 5983250
23.733 5932122
23.737 5906909.5
23.741 5934945
23.745 5949183
23.749 6085448
23.754 6052136.5
23.758 6067236.5
23.762 6031142.5
23.766 6024281.5
23.77 5940334.5
23.775 6010500.5
23.779 5972749
23.783 6059285
23.787 6028070
23.791 5974611.5
23.796 6024147
23.8 6017632.5
23.804 5974712
23.808 6014801.5
23.812 6046984
23.817 6048772.5
23.821 6139276
23.825 6174272
23.829 6249928.5
23.833 6268613
23.838 6243156.5
23.842 6271770.5
23.846 6304494.5
23.85 6263231
23.854 6202910
23.858 6267260
23.863 6259617.5
23.867 6380199.5
23.871 6416811.5
23.875 6357466
23.879 6356843
23.884 6284860.5
23.888 6265955.5
23.892 6299688.5
23.896 6258991.5
23.9 6313967.5
23.905 6290092
23.909 6239580
23.913 6176598
23.917 6198753
23.921 6193151
23.926 6203421
23.93 6187415
23.934 6245606.5
23.938 6172492
23.942 6197116.5
23.947 6120859.5
23.951 6045229
23.955 6100446.5
23.959 6120546
23.963 6073595
23.967 6048961
23.972 6100095
23.976 6232807
23.98 6200870.5
23.984 6134184.5
23.988 6198319.5
23.993 6242140.5
23.997 6249676.5
24.001 6301286.5
24.005 6313911.5
24.009 6333197.5
24.014 6326152
24.018 6295979.5
24.022 6294264.5
24.026 6273990.5
24.03 6236061.5
24.035 6214333
24.039 6247104.5
24.043 6268018.5
24.047 6318438.5
24.051 6319407.5
24.056 6440654.5
24.06 6413998
24.064 6401100
24.068 6328523.5
24.072 6396553
24.076 6341276
24.081 6326024
24.085 6299092
24.089 6300785
24.093 6356224.5
24.097 6335590.5
24.102 6299668.5
24.106 6328765
24.11 6406019
24.114 6305059
24.118 6306937
24.123 6326885.5
24.127 6375668.5
24.131 6392409
24.135 6320413.5
24.139 6270218.5
24.144 6306834.5
24.148 6302167.5
24.152 6364947.5
24.156 6423630
24.16 6398123
24.165 6404443.5
24.169 6457486
24.173 6467619
24.177 6464320.5
24.181 6475582.5
24.186 6514053.5
24.19 6528066.5
24.194 6507008
24.198 6408911
24.202 6283545
24.206 6329972.5
24.211 6273354
24.215 6327891.5
24.219 6408603
24.223 6382235
24.227 6382816.5
24.232 6386396
24.236 6392492.5
24.24 6328929.5
24.244 6313896.5
24.248 6332525
24.253 6304645.5
24.257 6305742.5
24.261 6411900
24.265 6451370.5
24.269 6456297.5
24.274 6577817
24.278 6618459.5
24.282 6703355
24.286 6670264.5
24.29 6677436
24.295 6669303.5
24.299 6695707.5
24.303 6799354.5
24.307 6753146.5
24.311 6736890
24.315 6660878
24.32 6535893.5
24.324 6511016
24.328 6644046
24.332 6633846.5
24.336 6648019
24.341 6706990.5
24.345 6617039
24.349 6668423
24.353 6559728
24.357 6540889
24.362 6561943
24.366 6549051.5
24.37 6604207.5
24.374 6608651.5
24.378 6630351.5
24.383 6625701
24.387 6716721
24.391 6692792
24.395 6598445
24.399 6514760.5
24.404 6527556
24.408 6569112.5
24.412 6633108
24.416 6653129
24.42 6689323.5
24.424 6720650
24.429 6743614.5
24.433 6810567.5
24.437 6767250
24.441 6719153
24.445 6853058.5
24.45 6907996.5
24.454 6951300
24.458 6952504
24.462 6995272.5
24.466 6961752.5
24.471 6995130.5
24.475 6911159.5
24.479 6920488
24.483 6893023.5
24.487 6802074.5
24.492 6854177.5
24.496 6829743.5
24.5 6859954
24.504 6810288.5
24.508 6903158.5
24.513 6938098.5
24.517 6961552
24.521 6961996.5
24.525 6902290.5
24.529 6857648
24.534 6926138
24.538 6925978
24.542 6869510
24.546 7005795
24.55 7067053.5
24.554 7164422.5
24.559 7288928
24.563 7476468.5
24.567 7666461
24.571 7734274.5
24.575 7647552.5
24.58 7616658.5
24.584 7530177.5
24.588 7443173
24.592 7333647
24.596 7304014
24.601 7271925.5
24.605 7218272.5
24.609 7219202
24.613 7208881.5
24.617 7107526.5
24.622 7006329.5
24.626 6984887.5
24.63 7058008.5
24.634 7099067.5
24.638 7147158
24.643 7200280.5
24.647 7216641.5
24.651 7194652.5
24.655 7177705.5
24.659 7207188
24.663 7228246.5
24.668 7178990
24.672 7147955
24.676 7169417
24.68 7340746
24.684 7482390
24.689 7625516.5
24.693 7858161.5
24.697 8051560.5
24.701 8169461.5
24.705 8270190.5
24.71 8390754
24.714 8491548
24.718 8485589
24.722 8489166
24.726 8308155
24.731 8212895.5
24.735 8059220.5
24.739 7914912.5
24.743 7841625.5
24.747 7731575
24.752 7579346.5
24.756 7599579
24.76 7628320
24.764 7836183
24.768 7978602
24.772 8000431.5
24.777 8158011
24.781 8357600
24.785 8534273
24.789 8710790
24.793 8760225
24.798 8785089
24.802 8697194
24.806 8509098
24.81 8428034
24.814 8344742.5
24.819 8278531
24.823 8131184.5
24.827 8018716
24.831 7851391.5
24.835 7758021
24.84 7844198
24.844 7703006.5
24.848 7802878.5
24.852 7690728
24.856 7717475
24.861 7786942.5
24.865 7721448
24.869 7623268.5
24.873 7664962.5
24.877 7656496.5
24.881 7760467
24.886 7670676
24.89 7750177.5
24.894 7703158
24.898 7765294.5
24.902 7853542
24.907 7922282
24.911 7854352.5
24.915 7912310
24.919 7825292.5
24.923 7890426
24.928 7800505
24.932 7852974
24.936 7816458
24.94 7668061
24.944 7656079
24.949 7659486.5
24.953 7702148
24.957 7732223
24.961 7654885.5
24.965 7681128
24.97 7779958.5
24.974 7782015
24.978 7739784
24.982 7715870.5
24.986 7644774
24.991 7670283.5
24.995 7690796
24.999 7688219
25.003 7699883.5
25.007 7787928
25.011 7806427
25.016 7764497.5
25.02 7911882
25.024 7816056.5
25.028 7751029
25.032 7793404
25.037 7756269
25.041 7752215
25.045 7799033
25.049 7755334
25.053 7789705.5
25.058 7762662
25.062 7713237
25.066 7725488.5
25.07 7809437
25.074 7838589.5
25.079 7846968.5
25.083 7833801.5
25.087 7849561.5
25.091 8020602
25.095 8006151
25.1 8061103
25.104 8037931.5
25.108 8002453.5
25.112 8018658.5
25.116 7970049
25.12 7969862.5
25.125 7945189.5
25.129 7941262
25.133 8004100
25.137 7968789.5
25.141 7928206.5
25.146 7925514.5
25.15 7964622
25.154 7950831.5
25.158 7935631
25.162 7944110
25.167 8036759.5
25.171 7902512.5
25.175 8073134.5
25.179 8149935.5
25.183 8148915
25.188 8154347.5
25.192 8111690
25.196 8030335
25.2 8153717.5
25.204 8143918.5
25.209 8028834.5
25.213 8030074.5
25.217 8038286.5
25.221 8109765
25.225 8128953.5
25.229 8097330
25.234 8011971
25.238 8069131
25.242 7979819
25.246 8024715.5
25.25 8021257.5
25.255 8023235
25.259 7989317.5
25.263 8079867.5
25.267 8081367.5
25.271 7965106.5
25.276 8035383
25.28 7976088.5
25.284 7972420.5
25.288 7962870.5
25.292 7936568
25.297 8007126.5
25.301 8078056
25.305 8052181.5
25.309 8016115.5
25.313 7997833.5
25.318 7955731
25.322 8093970.5
25.326 8041030
25.33 8128389
25.334 8164119
25.338 8236017.5
25.343 8211440
25.347 8369199
25.351 8510579
25.355 8626444
25.359 8709151
25.364 8666151
25.368 8745682
25.372 8687360
25.376 8622373
25.38 8497506
25.385 8368235.5
25.389 8317100.5
25.393 8207747
25.397 8197337
25.401 8140773.5
25.406 8219100
25.41 8135854
25.414 8297668
25.418 8252232.5
25.422 8305917
25.427 8166495
25.431 8152996
25.435 8180253.5
25.439 8201712
25.443 8250445.5
25.448 8170009
25.452 8212254.5
25.456 8157328
25.46 8215758.5
25.464 8096158
25.468 8210435
25.473 8236909.5
25.477 8218080
25.481 8163274
25.485 8122919.5
25.489 8155751.5
25.494 8087970
25.498 8156564.5
25.502 8189911.5
25.506 8170488
25.51 8135021.5
25.515 8070010.5
25.519 8151258.5
25.523 8162667.5
25.527 8102792
25.531 8114602
25.536 8151327.5
25.54 8187500
25.544 8101241.5
25.548 8089969.5
25.552 8116075.5
25.557 8171583.5
25.561 8186488.5
25.565 8245756.5
25.569 8256641.5
25.573 8288889.5
25.577 8233491
25.582 8281903.5
25.586 8277924
25.59 8317105
25.594 8330619.5
25.598 8224153.5
25.603 8243151
25.607 8266488
25.611 8295282.5
25.615 8309724.5
25.619 8250062.5
25.624 8281461.5
25.628 8314761
25.632 8287632
25.636 8328563
25.64 8322855.5
25.645 8358011.5
25.649 8335122.5
25.653 8310437.5
25.657 8265785.5
25.661 8436307
25.666 8451731
25.67 8444250
25.674 8364042.5
25.678 8324681.5
25.682 8318353.5
25.686 8343960
25.691 8367469
25.695 8308255.5
25.699 8273995
25.703 8320433.5
25.707 8430718
25.712 8307025
25.716 8331711
25.72 8449612
25.724 8417702
25.728 8435136
25.733 8426878
25.737 8399591
25.741 8369784.5
25.745 8345770.5
25.749 8353310.5
25.754 8337203
25.758 8444643
25.762 8472875
25.766 8410890
25.77 8332235.5
25.775 8374203.5
25.779 8430408
25.783 8379840
25.787 8315444
25.791 8297611
25.795 8320657.5
25.8 8308705.5
25.804 8346791.5
25.808 8384768
25.812 8283650.5
25.816 8277837.5
25.821 8272155.5
25.825 8255851.5
25.829 8211442.5
25.833 8174102.5
25.837 8183894.5
25.842 8259827
25.846 8276296.5
25.85 8302260
25.854 8350834.5
25.858 8309210
25.863 8303020
25.867 8304012.5
25.871 8374652
25.875 8419567
25.879 8392884
25.884 8307486.5
25.888 8309121.5
25.892 8331663.5
25.896 8409043
25.9 8339089
25.905 8331932.5
25.909 8370963
25.913 8274667
25.917 8303149
25.921 8248386
25.925 8216140.5
25.93 8337464.5
25.934 8257244.5
25.938 8300910.5
25.942 8361348
25.946 8303930
25.951 8260086
25.955 8346685
25.959 8268381.5
25.963 8255619.5
25.967 8283539
25.972 8363990.5
25.976 8346123.5
25.98 8374810
25.984 8281719
25.988 8322816
25.993 8240805.5
25.997 8283888.5
26.001 8235977.5
26.005 8298439
26.009 8304579
26.014 8365264.5
26.018 8287640.5
26.022 8329373
26.026 8313256.5
26.03 8367485.5
26.034 8310259
26.039 8255228
26.043 8317452
26.047 8312686.5
26.051 8241649
26.055 8230724.5
26.06 8286683.5
26.064 8208999
26.068 8202651
26.072 8293930
26.076 8248149
26.081 8244209
26.085 8231371.5
26.089 8372240
26.093 8377981.5
26.097 8311569
26.102 8366606
26.106 8410008
26.11 8332757
26.114 8347125
26.118 8305353
26.123 8264613
26.127 8265323.5
26.131 8360960
26.135 8493202
26.139 8511635
26.143 8552483
26.148 8503403
26.152 8694656
26.156 8993220
26.16 9174015
26.164 9384597
26.169 9539686
26.173 9770077
26.177 9923606
26.181 10225311
26.185 10392983
26.19 10532026
26.194 10558643
26.198 10501723
26.202 10464585
26.206 10310476
26.211 10158030
26.215 9978767
26.219 9900318
26.223 9646097
26.227 9414696
26.232 9259251
26.236 9220925
26.24 9035539
26.244 8966231
26.248 8962727
26.252 8804148
26.257 8685618
26.261 8651530
26.265 8598093
26.269 8608279
26.273 8634272
26.278 8627136
26.282 8611190
26.286 8543977
26.29 8513938
26.294 8491049
26.299 8545345
26.303 8549298
26.307 8541110
26.311 8635456
26.315 8601776
26.32 8523582
26.324 8552755
26.328 8513588
26.332 8543713
26.336 8486310
26.341 8416022
26.345 8478240
26.349 8555788
26.353 8515202
26.357 8423055
26.362 8423105
26.366 8350833.5
26.37 8387559.5
26.374 8361939
26.378 8452854
26.382 8441102
26.387 8450732
26.391 8472790
26.395 8473764
26.399 8494236
26.403 8415174
26.408 8486493
26.412 8570473
26.416 8370857
26.42 8429522
26.424 8446519
26.429 8394711
26.433 8496990
26.437 8356323.5
26.441 8356514
26.445 8249991
26.45 8348100
26.454 8295742
26.458 8300979
26.462 8344456
26.466 8332022
26.471 8345549.5
26.475 8349195
26.479 8368108.5
26.483 8410495
26.487 8415423
26.491 8507492
26.496 8555390
26.5 8619916
26.504 8899123
26.508 9314555
26.512 9819252
26.517 10308380
26.521 11078406
26.525 11775794
26.529 12477562
26.533 13235187
26.538 13831754
26.542 14041727
26.546 14243179
26.55 14064055
26.554 13754489
26.559 13297738
26.563 12713737
26.567 12244059
26.571 11698091
26.575 11141989
26.58 10675616
26.584 10336762
26.588 9997997
26.592 9658299
26.596 9391094
26.6 9221185
26.605 9047356
26.609 8938308
26.613 8864719
26.617 9019720
26.621 9004643
26.626 8950813
26.63 8885273
26.634 8840375
26.638 8849143
26.642 8833274
26.647 8907968
26.651 8981105
26.655 8908450
26.659 8948639
26.663 8798029
26.668 8773419
26.672 8622245
26.676 8627811
26.68 8582558
26.684 8611837
26.689 8683284
26.693 8784166
26.697 8608322
26.701 8562922
26.705 8552989
26.709 8551316
26.714 8528181
26.718 8465401
26.722 8511163
26.726 8444509
26.73 8487896
26.735 8463342
26.739 8557593
26.743 8539315
26.747 8544427
26.751 8530751
26.756 8497925
26.76 8591956
26.764 8512497
26.768 8545392
26.772 8419236
26.777 8433708
26.781 8426185
26.785 8503182
26.789 8489463
26.793 8448756
26.798 8470058
26.802 8306736.5
26.806 8391039
26.81 8399886
26.814 8377934.5
26.818 8325751.5
26.823 8384334.5
26.827 8337870.5
26.831 8356743.5
26.835 8358501
26.839 8395194
26.844 8465512
26.848 8335338
26.852 8456975
26.856 8394135
26.86 8275564
26.865 8376222.5
26.869 8262854
26.873 8281708.5
26.877 8281382.5
26.881 8318996.5
26.886 8261345.5
26.89 8322988.5
26.894 8343999.5
26.898 8408895
26.902 8410372
26.907 8277407
26.911 8243853.5
26.915 8275238
26.919 8441220
26.923 8340394
26.928 8384075.5
26.932 8382513.5
26.936 8285202.5
26.94 8228859.5
26.944 8148044.5
26.948 8179537.5
26.953 8172819.5
26.957 8294359.5
26.961 8201913
26.965 8285174
26.969 8318461.5
26.974 8254042
26.978 8209535
26.982 8289180
26.986 8292737.5
26.99 8238964.5
26.995 8344079
26.999 8307258.5
27.003 8292730.5
27.007 8355749.5
27.011 8350515.5
27.016 8230180
27.02 8322320
27.024 8274944.5
27.028 8297724
27.032 8404326
27.037 8353324
27.041 8450772
27.045 8369091
27.049 8437076
27.053 8432804
27.057 8386714
27.062 8220540.5
27.066 8293806.5
27.07 8358308
27.074 8194687.5
27.078 8187110
27.083 8326029.5
27.087 8332938
27.091 8337573
27.095 8325239
27.099 8375757.5
27.104 8359217.5
27.108 8287970
27.112 8395822
27.116 8422747
27.12 8436252
27.125 8337262.5
27.129 8391449
27.133 8425327
27.137 8430295
27.141 8324735
27.146 8383773
27.15 8468543
27.154 8698433
27.158 9115559
27.162 9424252
27.166 9972399
27.171 10838220
27.175 11861847
27.179 13233903
27.183 14954183
27.187 17073662
27.192 19124148
27.196 21170700
27.2 23336588
27.204 25459602
27.208 27400636
27.213 28825540
27.217 29289576
27.221 29147682
27.225 28598218
27.229 27538014
27.234 26112328
27.238 23897814
27.242 21610134
27.246 19746668
27.25 17987548
27.255 16282669
27.259 14992142
27.263 13920262
27.267 13030023
27.271 12234679
27.275 11634159
27.28 11256381
27.284 10955229
27.288 10609058
27.292 10371175
27.296 10293098
27.301 10232087
27.305 10115856
27.309 10043848
27.313 9890066
27.317 9908626
27.322 9968333
27.326 9957338
27.33 9928308
27.334 10032049
27.338 10017433
27.343 10037156
27.347 9988031
27.351 10057622
27.355 9882791
27.359 9781554
27.364 9765169
27.368 9676673
27.372 9599888
27.376 9629839
27.38 9547301
27.384 9421922
27.389 9487892
27.393 9606323
27.397 9598769
27.401 9664692
27.405 9742866
27.41 9818071
27.414 9942856
27.418 9807448
27.422 9976882
27.426 9977571
27.431 9948242
27.435 9852267
27.439 9862496
27.443 9671618
27.447 9454615
27.452 9251730
27.456 9248388
27.46 9193793
27.464 9035876
27.468 8995256
27.473 8928054
27.477 8907400
27.481 8832358
27.485 8753001
27.489 8644210
27.494 8626264
27.498 8688219
27.502 8582148
27.506 8606420
27.51 8591248
27.514 8654278
27.519 8605830
27.523 8599958
27.527 8595010
27.531 8570306
27.535 8594942
27.54 8494686
27.544 8470439
27.548 8538961
27.552 8514074
27.556 8453701
27.561 8441610
27.565 8422511
27.569 8430883
27.573 8541990
27.577 8492775
27.582 8482832
27.586 8462707
27.59 8429200
27.594 8562639
27.598 8519622
27.603 8416643
27.607 8292512
27.611 8219558.5
27.615 8282259
27.619 8431650
27.623 8543683
27.628 8492101
27.632 8416502
27.636 8464140
27.64 8440092
27.644 8551835
27.649 8534106
27.653 8630110
27.657 8716796
27.661 8550476
27.665 8630795
27.67 8612801
27.674 8672250
27.678 8625368
27.682 8689908
27.686 8703053
27.691 8724479
27.695 8673314
27.699 8601794
27.703 8543143
27.707 8604476
27.712 8566289
27.716 8487495
27.72 8525052
27.724 8377650
27.728 8410463
27.732 8475250
27.737 8606248
27.741 8632853
27.745 8603119
27.749 8569585
27.753 8587244
27.758 8510132
27.762 8511449
27.766 8530426
27.77 8541145
27.774 8639938
27.779 8619448
27.783 8576722
27.787 8426290
27.791 8465356
27.795 8337405
27.8 8461566
27.804 8285262.5
27.808 8357891.5
27.812 8308211.5
27.816 8291900.5
27.821 8198821.5
27.825 8192866
27.829 8326828.5
27.833 8356402
27.837 8436542
27.841 8484171
27.846 8333643.5
27.85 8312170.5
27.854 8369111.5
27.858 8547882
27.862 8636397
27.867 8543718
27.871 8505340
27.875 8496345
27.879 8578513
27.883 8569508
27.888 8621389
27.892 8528321
27.896 8478668
27.9 8475224
27.904 8541022
27.909 8332300.5
27.913 8346101
27.917 8370939.5
27.921 8451038
27.925 8457561
27.93 8376481
27.934 8342731
27.938 8282102.5
27.942 8235803.5
27.946 8342358
27.951 8373661
27.955 8365918.5
27.959 8295399.5
27.963 8161837
27.967 8174181.5
27.971 8184519.5
27.976 8216131
27.98 8115205.5
27.984 8212010.5
27.988 8173823.5
27.992 8237330
27.997 8254387
28.001 8274234
28.005 8246650
28.009 8353169.5
28.013 8259682.5
28.018 8239270.5
28.022 8240972
28.026 8236170.5
28.03 8320082
28.034 8338169.5
28.039 8338414
28.043 8326767.5
28.047 8230610
28.051 8250542.5
28.055 8287641
28.06 8270297.5
28.064 8162611
28.068 8185004
28.072 8240208.5
28.076 8158679
28.08 8099035.5
28.085 8077431
28.089 8110561
28.093 8263616.5
28.097 8172098.5
28.101 8170649
28.106 8268690
28.11 8390806
28.114 8433229
28.118 8327464.5
28.122 8270010
28.127 8232698
28.131 8272928.5
28.135 8253198.5
28.139 8175394.5
28.143 8234882.5
28.148 8360832.5
28.152 8541533
28.156 8533992
28.16 8609071
28.164 8717866
28.169 8750077
28.173 8761946
28.177 8559318
28.181 8630041
28.185 8646401
28.189 8666404
28.194 8660292
28.198 8666712
28.202 8638450
28.206 8573134
28.21 8551106
28.215 8529273
28.219 8473880
28.223 8401332
28.227 8340431
28.231 8340947
28.236 8364404
28.24 8288091
28.244 8297653
28.248 8301301
28.252 8269283
28.257 8226546
28.261 8239904
28.265 8287527.5
28.269 8263462.5
28.273 8229029.5
28.278 8175507.5
28.282 8095466
28.286 8190782.5
28.29 8158553.5
28.294 8211779
28.298 8122361.5
28.303 8194608
28.307 8176539
28.311 8216028.5
28.315 8276328
28.319 8235364
28.324 8163153.5
28.328 8112820.5
28.332 8032280
28.336 8265923
28.34 8107437.5
28.345 8083887
28.349 8074480.5
28.353 8081032.5
28.357 8139403.5
28.361 8155717
28.366 8097307
28.37 8118491.5
28.374 8109523
28.378 8066727.5
28.382 8124785.5
28.387 8157296.5
28.391 8179954
28.395 8199248
28.399 8211289.5
28.403 8214766
28.408 8191294
28.412 8184697.5
28.416 8243860
28.42 8284529.5
28.424 8307835
28.428 8265536
28.433 8226305.5
28.437 8278015.5
28.441 8357245.5
28.445 8404885
28.449 8469164
28.454 8555872
28.458 8422931
28.462 8370893.5
28.466 8320943.5
28.47 8272476
28.475 8308906.5
28.479 8279477
28.483 8352562.5
28.487 8235026.5
28.491 8232576.5
28.496 8265670.5
28.5 8283227.5
28.504 8376947
28.508 8357708.5
28.512 8290189.5
28.517 8399692
28.521 8352415
28.525 8365943.5
28.529 8396154
28.533 8347320
28.537 8320957
28.542 8277296.5
28.546 8289781
28.55 8151039.5
28.554 8203583.5
28.558 8168246.5
28.563 8176202
28.567 8188733.5
28.571 8286388
28.575 8180566
28.579 8256844.5
28.584 8191419.5
28.588 8169370.5
28.592 8044740
28.596 8052364.5
28.6 8118213.5
28.605 8031662.5
28.609 8181897.5
28.613 8115716
28.617 8070768
28.621 8011495
28.626 7981531
28.63 8024638
28.634 8066089.5
28.638 8018587
28.642 7951027
28.646 7993246
28.651 7988124
28.655 8051226
28.659 8150619
28.663 8113501.5
28.667 8104100.5
28.672 8017132
28.676 7967871
28.68 7949061
28.684 8106994.5
28.688 8065451.5
28.693 8003550
28.697 7930559.5
28.701 7940151
28.705 7897738
28.709 8005798
28.714 8032137.5
28.718 7988199.5
28.722 7949345
28.726 8055058.5
28.73 7995442.5
28.735 8029789
28.739 8034465
28.743 8116965.5
28.747 8027064
28.751 7995619.5
28.755 8063267
28.76 8120902
28.764 8051498
28.768 8046212.5
28.772 8108092
28.776 8037702
28.781 8012972
28.785 7924835
28.789 8032773.5
28.793 8109380
28.797 8088110.5
28.802 8051401
28.806 7995629
28.81 7959854
28.814 7982278.5
28.818 8048564
28.823 8064760.5
28.827 8038739
28.831 7970670.5
28.835 8004751.5
28.839 8117776
28.844 8017113
28.848 7900920
28.852 7974682
28.856 8027791
28.86 8077417.5
28.865 8050757
28.869 8078557.5
28.873 8102334
28.877 8137338.5
28.881 8079876.5
28.885 7983801
28.89 8022362
28.894 7994959.5
28.898 7952421.5
28.902 7999850
28.906 8119603
28.911 8007656.5
28.915 8045685
28.919 8057184.5
28.923 8073156.5
28.927 8019272
28.932 8066591.5
28.936 8053266
28.94 8055047
28.944 8116326
28.948 8237909.5
28.953 8340406
28.957 8289747.5
28.961 8409137
28.965 8605298
28.969 8730194
28.974 9011990
28.978 9209744
28.982 9358115
28.986 9692729
28.99 10019774
28.994 10240366
28.999 10487214
29.003 10764238
29.007 10986919
29.011 11156807
29.015 11037290
29.02 10891594
29.024 10725374
29.028 10497300
29.032 10230169
29.036 10042342
29.041 9865223
29.045 9693602
29.049 9289809
29.053 9188766
29.057 8990076
29.062 8904150
29.066 8853268
29.07 8764266
29.074 8616895
29.078 8624858
29.083 8554364
29.087 8493897
29.091 8458115
29.095 8315937
29.099 8366229
29.103 8313772.5
29.108 8308461
29.112 8171523
29.116 8155663
29.12 8139229.5
29.124 8222380.5
29.129 8248162
29.133 8230711
29.137 8303357
29.141 8118409.5
29.145 8172298
29.15 8178449
29.154 8154724.5
29.158 8130145
29.162 8082539.5
29.166 8123006
29.171 8097345.5
29.175 8028394
29.179 8042298.5
29.183 8071869.5
29.187 7997658.5
29.192 8024943
29.196 8080767
29.2 8134624.5
29.204 8098052
29.208 8078616
29.212 8062078
29.217 8069986
29.221 8133057.5
29.225 8178135.5
29.229 8119975.5
29.233 8055633.5
29.238 8156833.5
29.242 8057708
29.246 8190472.5
29.25 8148734
29.254 8274554
29.259 8369593.5
29.263 8278094
29.267 8194008
29.271 8260986.5
29.275 8409764
29.28 8320473
29.284 8265169
29.288 8313458.5
29.292 8280637.5
29.296 8299818.5
29.301 8258913
29.305 8250723.5
29.309 8186362
29.313 8278835
29.317 8286229.5
29.322 8194806.5
29.326 8211236
29.33 8251455
29.334 8268707.5
29.338 8157648.5
29.342 8110537
29.347 8071199.5
29.351 8150489.5
29.355 8241626.5
29.359 8212872.5
29.363 8172376
29.368 8161618.5
29.372 8151718
29.376 8107532.5
29.38 8004542.5
29.384 8077276
29.389 8104531.5
29.393 8093103.5
29.397 7974510
29.401 8101176.5
29.405 8132283
29.41 8194224
29.414 8098574.5
29.418 8178309.5
29.422 8167595.5
29.426 8117107
29.431 8184033
29.435 8205290
29.439 8233638
29.443 8247678.5
29.447 8200440
29.451 8129008.5
29.456 8099177.5
29.46 8196722
29.464 8251411
29.468 8122565.5
29.472 8222385.5
29.477 8261507.5
29.481 8187861.5
29.485 8220609.5
29.489 8159041
29.493 8082719.5
29.498 8103915.5
29.502 8064674
29.506 8098534
29.51 8071659
29.514 8026380.5
29.519 7955741.5
29.523 7908011
29.527 8000034
29.531 8076397.5
29.535 8108170
29.54 8055018
29.544 8081262
29.548 8121295.5
29.552 8119573
29.556 8099490.5
29.56 8094536.5
29.565 8016643.5
29.569 8044473.5
29.573 7956870.5
29.577 7865486
29.581 7982534
29.586 7877595
29.59 7930874
29.594 7990280.5
29.598 8002294
29.602 8037557
29.607 7927564
29.611 7861449.5
29.615 7864527.5
29.619 7894202.5
29.623 7974717.5
29.628 8080125.5
29.632 7932986
29.636 7941781
29.64 8050457
29.644 7877382.5
29.649 8007034.5
29.653 7935098
29.657 7949311.5
29.661 7896218
29.665 7937340.5
29.669 7973715
29.674 8007233
29.678 8074282.5
29.682 8061665.5
29.686 7961299.5
29.69 8015814
29.695 8126642.5
29.699 8041453
29.703 7993652.5
29.707 8084716.5
29.711 8057030
29.716 7999082
29.72 7886619.5
29.724 7897494.5
29.728 8000509
29.732 7997768.5
29.737 7945547
29.741 8020324
29.745 8037133.5
29.749 8066937.5
29.753 7984683
29.758 7946944
29.762 7968516
29.766 7999936.5
29.77 7964693.5
29.774 8058687.5
29.778 8112815
29.783 8021142.5
29.787 7972902
29.791 7880588.5
29.795 7959172.5
29.799 8015934
29.804 8018593
29.808 8002575.5
29.812 7979752
29.816 7980289.5
29.82 8032050
29.825 7979545.5
29.829 7899045
29.833 8013810.5
29.837 7948697.5
29.841 7840793
29.846 7924789
29.85 7955332.5
29.854 7907474
29.858 7896999
29.862 7932103.5
29.867 8022123
29.871 8052158.5
29.875 7982772.5
29.879 7946697.5
29.883 7968074
29.888 7924588
29.892 8012066
29.896 8003513.5
29.9 8018699.5
29.904 7967713
29.908 8017019
29.913 7963279
29.917 7971568
29.921 7998826
29.925 8002471
29.929 7855632.5
29.934 7845121.5
29.938 7864029.5
29.942 7824796.5
29.946 7897842
29.95 7946787
29.955 7972857.5
29.959 7984383
29.963 8011022
29.967 8000622
29.971 8001548
29.976 8012484
29.98 7925796
29.984 7946921
29.988 8037098.5
29.992 8344855.5
29.997 10275575
30.001 12057197

Round 2
Reviewer 2 Report
Dear authors,
I still have my problem with yours data presentation. Please read about Kovats indices in gas-chromatography.
For example nonadecane has an Kovats index of 1900 by definition. Octadecanoic acid has an index of 2170 while 1-Hexadecanolacetate is around 2010 and cannot elute behind Octadecanoic acid.
Therefor your sequence of eluted compounds in Table 1 is not possible and the assignment, at least for the sample HA, is still wrong!
My recommendation is to calibrate the column with a test mixture of aliphates. Exchange the retention time in the table by the retention index.
In your software you can select the use of Kovats indices.
The assignment of branched aliphatic compounds is not possible in every case.
Thus the columns of the HA sample may be more or less empty. You can look in the NIST Web database if for yours compounds are RI value known.
You can assign Kovats indices to yours phenylethylchromones.
You forgot to describe yours MS/MS mass spectrometer in the material and methods section. Agilent 7890B is only the gas chromatograph.
Author Response
Thank you for your correction. I'm sorry for this mistake. All the hints and remarks have been carefully checked and considered in our revised “manuscript”, and the modifications are highlighted in yellow color.
Question: I still have my problem with yours data presentation. Please read about Kovats indices in gas-chromatography. For example nonadecane has an Kovats index of 1900 by definition. Octadecanoic acid has an index of 2170 while 1-Hexadecanolacetate is around 2010 and cannot elute behind Octadecanoic acid. Therefor your sequence of eluted compounds in Table 1 is not possible and the assignment, at least for the sample HA, is still wrong!
My recommendation is to calibrate the column with a test mixture of aliphates. Exchange the retention time in the table by the retention index. In your software you can select the use of Kovats indices.The assignment of branched aliphatic compounds is not possible in every case.Thus the columns of the HA sample may be more or less empty. You can look in the NIST Web database if for yours compounds are RI value known. You can assign Kovats indices to yours phenylethylchromones.You forgot to describe yours MS/MS mass spectrometer in the material and methods section. Agilent 7890B is only the gas chromatograph.
Response: Dear reviewer, we appreciate your valuable suggestions. The MS/MS mass spectrometer is described on line 319. Owing this study is mainly about the comparison of biological activities of wild agarwood and induced agarwood, HA is not so important in the text, so we decided to delete the data and related content about HA. In addition, we have revised the contents of lines 197-200, and line 305.
